There are amendments to this paper

# Spatially clustered loci with multiple enhancers are frequent targets of HIV-1 integration

Bojana Lucic[1,11], Heng-Chang Chen [2,3,11], Maja Kuzman [4,11], Eduard Zorita [2,3,11], Julia Wegner[1,10], Vera Minneker[5], Wei Wang[6], Raffaele Fronza[6], Stefanie Laufs[6], Manfred Schmidt[6], Ralph Stadhouders[7,8], Vassilis Roukos [5], Kristian Vlahovicek [4], Guillaume J. Filion [2,3,9] & Marina Lusic [1]

HIV-1 recurrently targets active genes and integrates in the proximity of the nuclear pore compartment in CD4$^+$ T cells. However, the genomic features of these genes and the relevance of their transcriptional activity for HIV-1 integration have so far remained unclear. Here we show that recurrently targeted genes are proximal to super-enhancer genomic elements and that they cluster in specific spatial compartments of the T cell nucleus. We further show that these gene clusters acquire their location during the activation of T cells. The clustering of these genes along with their transcriptional activity are the major determinants of HIV-1 integration in T cells. Our results provide evidence of the relevance of the spatial compartmentalization of the genome for HIV-1 integration, thus further strengthening the role of nuclear architecture in viral infection.

[1] Department of Infectious Diseases, Integrative Virology, Heidelberg University Hospital and German Center for Infection Research, Heidelberg, Germany. [2] Genome Architecture, Gene Regulation, Stem Cells and Cancer Programme, Center for Genomic Regulation (CRG), Barcelona Institute of Science and Technology, Barcelona, Spain. [3] University Pompeu Fabra, Barcelona, Spain. [4] Bioinformatics Group, Division of Molecular Biology, Department of Biology, Faculty of Science, University of Zagreb, Zagreb, Croatia. [5] Institute of Molecular Biology (IMB), Mainz, Germany. [6] German Cancer Research Center (DKFZ) and National Center for Tumor Diseases (NCT), Heidelberg, Germany. [7] Department of Pulmonary Medicine, Erasmus MC, Rotterdam, The Netherlands. [8] Department of Cell Biology, Erasmus MC, Rotterdam, The Netherlands. [9] Department of Biological Sciences, University of Toronto Scarborough, Toronto, ON, Canada. [10] Present address: Institute for Clinical Chemistry and Clinical Pharmacology, Universitätsklinikum Bonn, Bonn, Germany. [11] These authors contributed equally: Bojana Lucic, Heng-Chang Chen, Maja Kuzman, Eduard Zorita. [12] These authors jointly supervised this work: Kristian Vlahovicek, Guillaume J. Filion, Marina Lusic. Correspondence and requests for materials should be addressed to K.V. (email: kristian@bioinfo.hr) or to G.J.F. (email: guillaume.filion@gmail.com) or to M.L. (email: Marina.lusic@med.uni-heidelberg.de)

ntegration of the proviral genome into the host chromosomal DNA is one of the defining features of retroviral replication[1–3]. Following integration, the viral genome can either be expressed or enter a transcriptionally dormant stage, establishing a reservoir of latently infected cells. Latently infected cells are indistinguishable from the non-infected ones and are therefore not eliminated by immune clearance mechanisms or recognized by current antiretroviral treatments[4,5]. Resting CD4[+] T cells of the memory phenotype represent the main reservoir of latent human immunodeficiency virus type 1 (HIV-1)[6]. However, it is still unclear how these reservoirs are established, as HIV-1 does not efficiently infect resting T cells due to different blocks at both pre-integration and integration levels[4,7–10]. One possible explanation is that some of the activated CD4[+] T cells revert back to the resting state upon infection with HIV-1, generating the reservoirs of silenced but replication-competent viruses[4]. What remains still to be defined is how this transition from activated to resting state occurs, and what changes in the cellular genome and chromatin are involved[11,12].

In activated CD4[+] T cells, the viral DNA enters the nucleus to access chromatin[13] passing through the nuclear pore complex (NPC)[14–16]. Nuclear pore proteins are important factors for the viral nuclear entry[17], as well as for the positioning and consequent integration of the viral DNA into the cellular genome[3,13–16,18,19]. Integration is not a random process, as HIV-1 predominantly integrates into active genes in gene-dense regions[20], mediated by the action of viral proteins integrase (IN) and capsid (CA). Through its interaction with LEDGF/p75[21–23], IN guides the integration into gene bodies. This pattern is shifted toward 5′ end regions of genes[22,24,25] or toward gene-poor regions[25] when LEDGF/p75 is depleted. Through its interaction with cleavage and polyadenylation specificity factor 6 (CPSF6), HIV-1 CA also contributes to the location of the viral genome[24,26,27]. Lack of CPSF6 arrests the incoming viral particles at the level of the NPC[27] or retargets the integrating viral DNA to the lamina-associated heterochromatin domains[26].

It is well established that HIV-1 targets open chromatin regions of active transcription and regions bearing enhancer marks[20,28,29]. Unlike typical enhancers, genomic elements known as super-enhancers (SEs) are defined by high levels of acetylated lysine 27 of histone 3 (H3K27ac) and binding of transcriptional co-activators, such as bromodomain-containing protein 4 (BRD4), the mediator complex[30], and the p300 histone acetyltransferase[30–32]. SEs control the expression of genes that define cell identity[30,32–34], and in case of CD4[+] T cells, relevant for HIV-1 infection, they control cytokines, cytokine receptors, and transcription factors regulating T cell-specific transcriptional programs[35]. Strikingly, one of the strongest immune-activation SEs[36,37] encoding for transcription factor BACH2 is among the most frequently targeted HIV-1 integration genes[38,39]. SE elements of cell identity genes were shown to be bound by nuclear pore proteins, which regulate their expression[40,41] and anchor them to the nuclear periphery[41]. Moreover, SEs seem to play a general role in organizing the genome through higher-order chromatin structures and architectural chromatin loops[42–44].

Evidence accumulated in the past decade has revealed that the chromosomal contacts, achieved by genome folding and looping, define separate compartments in the nucleus[45]. Hi-C data have shown that transcribed genes make preferential contacts with other transcribed genes, forming a spatial cluster known as the A compartment[46,47]. Reciprocally, silent genes and intergenic regions form a spatial cluster known as the B compartment. The loci of the B compartment are usually in contact with the nuclear lamina[48], i.e., at the periphery of the nucleus, where low levels of gene expression and heterochromatin histone signatures are found. In fact, these regions are almost completely avoided by HIV-1[18,25,26], whereas HIV-1 targets regions of open chromatin, which in some studies map in proximity to the NPC[18,19,49].

This suggests that a complex and dynamic interplay between the incoming virus, the host cell chromatin, and the dynamic nuclear organization contribute to the selection of genomic sequences into which HIV-1 integrates.

Here we find that HIV-1 integrates in proximity of SEs in patients and in T cell cultures in vitro. The observed phenomenon does not depend on the activity of SEs but on their position in spatial neighborhoods where HIV-1 insertion is facilitated. Consistently, HIV-1 integration hotspots cluster in the nuclear space and tend to contact SEs. Finally, we find that SE activity is critical to reorganize the genome of activated T cells, showing that they indirectly contribute to HIV-1 insertion biases.

## Results

**HIV-1 integrates in genes proximal to SEs.** We assembled a list of 4031 HIV-1 integration sites from activated primary CD4[+] T cells infected in vitro (ref. [50] and this study) and 9519 insertion sites from 6 studies from HIV-1 patients[38,39,51–54] (Supplementary Table 1). Ten thousand seven hundred and thirty-five integrations were in gene bodies (77% averaged over patient studies and 84% over in vitro infection studies), targeting a total of 5601 different genes (Supplementary Fig. 1a). This insertion dataset is not saturating (Supplementary Fig. 1b), yet we found that a subset of genes are recurrent HIV-1 targets, consistent with our previous findings[18]. We thus defined recurrent integration genes (RIGs) as genes with ≥1 HIV-1 integrations in at least 2 out of 8 datasets (see "Methods"), yielding a total of 1648 RIGs (Supplementary Fig. 1c).

To characterize RIGs, we extracted protein-coding genes without HIV-1 insertions in any dataset (called non-RIGs in the analysis, consisting of 13,140 genes) and compared their chromatin immunoprecipitation sequencing (ChIP-Seq) features in primary CD4[+] T cells. We first analyzed the levels of epigenomic features on protein-coding genes (Fig. 1a). As previously reported[18,50], we observed higher levels of H3K27ac, H3K4me1, and H3K4me3, as well as BRD4 and mediator of RNA polymerase II transcription subunit 1 (MED1) at transcription start sites of RIGs vs non-RIGs. Histone profiles of H3K36me3 and H4K20me1 were higher throughout RIG gene bodies, while the repressive transcription mark H3K27me3 was lower on RIGs vs non-RIGs. Of note, the mark of facultative heterochromatin H3K9me2 was depleted at transcription start sites of RIGs but remained unchanged throughout the gene body of RIGs vs non-RIGs.

In order to test the specificity of chromatin signatures of HIV-1 integration sites, we adapted the receiver operating characteristic (ROC) analysis[55,56]. We used control sites matched according to the distance to the nearest gene (see "Methods") and confirmed significant enrichment of the following genomic features: H3K27ac, H3K4me1, BRD4, MED1, H3K36me3, and H4K20me1 (Fig. 1b). The marks H3K27ac, H3K4me1, and H3K36me3, characteristic of active enhancers[57], cell type-specific enhancers[58], and bodies of transcribed genes[59], respectively, were the most enriched in the proximity of insertion sites. Consistent with the presence of H3K27ac and H3K4me1, we also found significant enrichment of BRD4, a constituent of SE genomic elements[30,32] (Fig. 1b). On average, 60% of insertion sites were significantly enriched in these chromatin marks (not shown) while we observed depletion of H3K27me3 and H3K9me2 in the proximity of insertion sites. Interestingly, we did not observe a statistically significant enrichment of H3K4me3 in the proximity of insertion sites.

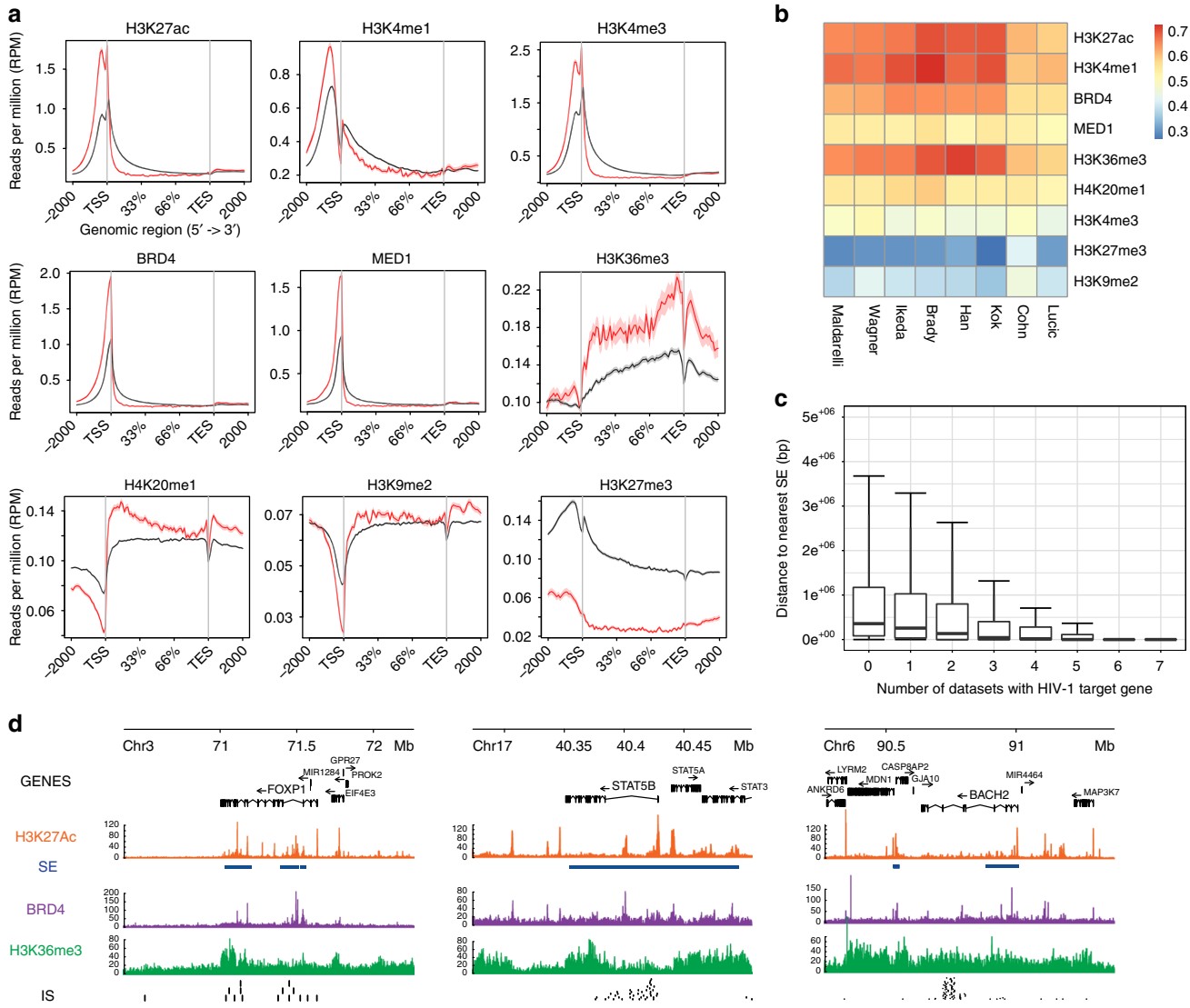

**Fig. 1** HIV-1 integration hotspots are within genes proximal to super-enhancers (SEs). **a** Metagene plots of H3K27ac, H3K4me1, H3K4me3, BRD4, MED1, H3K36me3, H4K20me1, H3K9me2, and H3K27me3 ChIP-Seq signals in recurrent integration genes (RIGs), which are protein coding in red and the rest of the protein-coding genes that are not targeted by HIV-1 (no RIGs) in black. **b** ROC analysis represented in heatmap summarizing the co-occurrence of integration sites and epigenetic modification obtained by ChIP-Seq for H3K27ac, H3K4me1, BRD4, MED1, H3K36me3, H4K20me1, H3K4me3, H3K27me3, and H3K9me2. HIV-1 integration datasets are shown in the columns, and epigenetic modifications are shown in rows. Associations are quantified using the ROC area method; values of ROC areas are shown in the color key at the right. **c** Distance to the nearest SE in activated CD4$^+$ T cells. Box plots represent distances from the gene to the nearest SE grouped by number of times the gene is found in different datasets. **d** *FOXP1*, *STAT5B*, and *BACH2* IS (black) superimposition on H3K27ac (orange), SE (blue), H3K36me3 (green), and BRD4 (violet) ChIP-Seq tracks

To confirm these trends, we identified SEs in activated CD4$^+$ T cells using H3K27ac ChIP-Seq and merged them with the SEs in activated CD4$^+$ T cells from dbSuper[60,61]. We obtained 2584 SEs, intersecting 564 RIGs (34.22%, Supplementary Fig. 1d). In addition, the more a RIG is targeted by HIV-1 (i.e., the higher the number of datasets where HIV-1 insertions are found in the gene), the closer it lies to SEs on average (Fig. 1c). In contrast, the insertion sites of the retrovirus HTLV-1[62] (human T lymphotropic virus type 1) were not enriched in SE marks (Supplementary Fig. 1e), while murine leukemia virus (MLV) showed a strong enrichment in all SE marks as expected[63]. Figure 1d shows the integration biases at gene scale on *FOXP1*, *STAT5B*, and *BACH2*, three highly targeted RIGs involved in T cell differentiation and activity. The ChIP-Seq profiles of H3K27ac, H3K36me3, and BRD4 indicate prominent clustering of HIV-1 insertion sites near the SEs defined by those marks. Thus HIV-1 displays specific preference to integrate into genes

proximal to SEs, herein defined as genomic elements of retroviral integrations.

**RIGs are proximal to SEs regardless of their expression**. HIV-1 is known to integrate into highly expressed genes[20,29]. It is thus possible that genes with an SE are targeted more often because they are expressed at a higher level. To test whether this is the case, we measured the transcript abundance of protein-coding genes in CD3/CD28-activated CD4$^+$ T cells by RNA sequencing (RNA-Seq). The mean expression of genes with HIV-1 insertions is higher than those not targeted by HIV-1 (Fig. 2a). More specifically, 21.4% of protein-coding genes targeted by HIV-1 are in the top 10% most expressed genes, compared to 6.07% of non-targeted genes. Moreover, the genes more often targeted by HIV-1 (RIGs) are expressed at higher levels (Fig. 2b), thus confirming that HIV-1 is biased toward highly expressed genes.

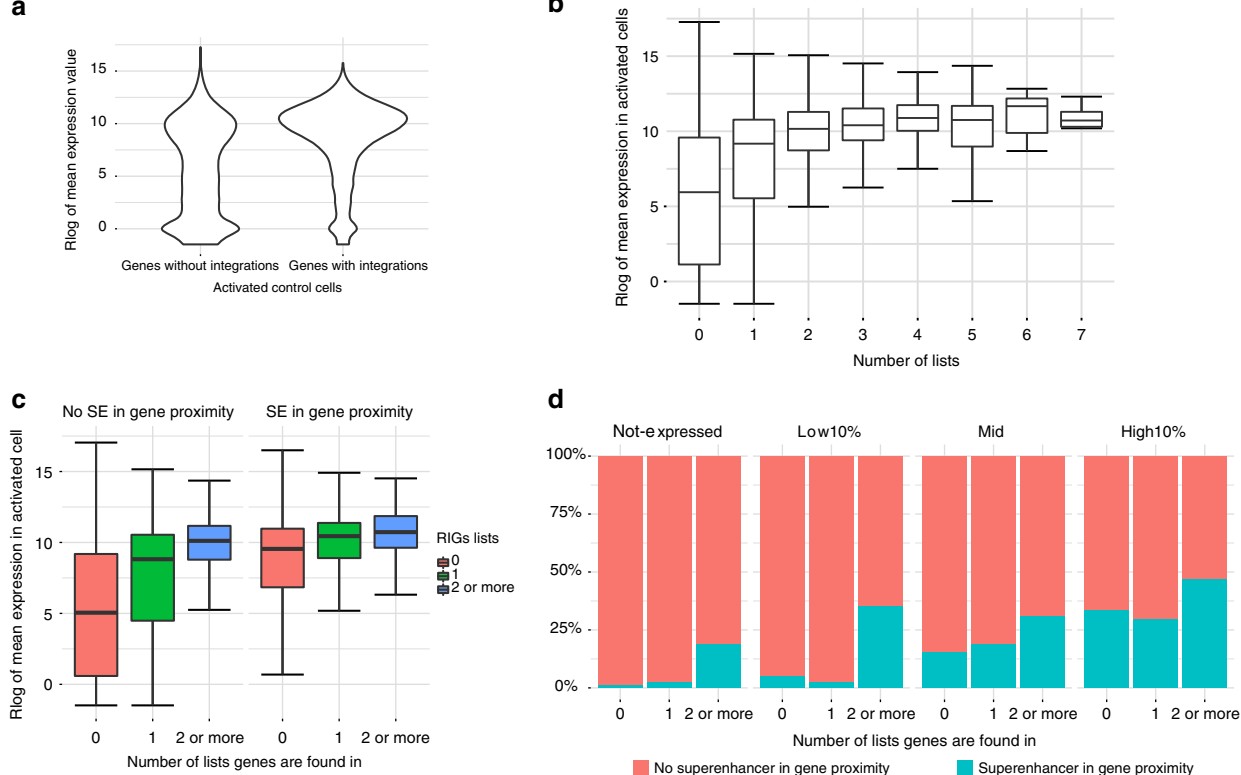

**Fig. 2** RIGs are proximal to super-enhancers regardless of their expression. **a** Regularized log-transformed read counts on protein-coding genes averaged over three replicates in activated CD4$^+$ T cells shown as violin plot for genes without HIV-1 integrations and genes with HIV-1 integrations. **b** Box plot for protein-coding genes grouped by number of HIV-1 lists they appear in. **c** Box plot for protein-coding genes grouped by the number of HIV-1 lists they appear in, with RIGs grouped together in ≥2 lists' group. Box plots are shown separately for genes that have super-enhancer 5 kb upstream of TSS or super-enhancer overlaps them (SE in proximity) and genes that do not have super-enhancer in proximity. Differences in median abundances of mRNA are statistically significant for all groups (p value <2.2 x 10$^{-16}$ for genes without HIV integrations and genes found on only one list and p value 3.7 × 10$^{-12}$ for RIGs, calculated by Wilcoxon rank-sum test). **d** Bar plots show the percentage of protein-coding genes that have super-enhancer in proximity, arranged by number of lists the gene is found in and by expression group

On average, genes with a SE are expressed at higher levels than those without (Fig. 2c). This trend is more subtle for RIGs, as they are expressed at a high level, with or without SEs (Fig. 2c, compare the blue boxes). However, RIGs are more often in the proximity of SEs than non-RIGs, irrespective of their expression (Fig. 2d). In particular, 19.05% of RIGs that are silent also have a proximal SE, while this is true for only 1.5% of the silent genes that were never found to be HIV-1 targets (Fig. 2d, leftmost panel). The trend remains the same for expressed genes (Fig. 2d) after dividing them into "low," "medium," and "high" expression groups (see "Methods"). In summary, our gene expression analysis suggests that genes recurrently targeted by HIV-1 have adjacent SE elements, irrespective of their transcriptional levels.

We next assessed the relationship between HIV-1 integration and transcription of genes controlled by SEs by using JQ1, a bromodomain and extraterminal domain protein inhibitor that prevents BRD4 binding to acetylated chromatin[64] and causes a subsequent dysregulation of RNA Pol II binding[31].

*MYC* is known to be regulated by SEs[31], so we used the *MYC* RNA and protein levels as a control for the JQ1 treatment in CD4$^+$ T cells (Supplementary Fig. 2a). We compared the HIV-1 insertion profiles with or without JQ1 by inverse PCR (see "Methods"). We mapped a total of 38,964 HIV-1 insertion sites and did not observe, at the chromosome scale, that JQ1 affects the insertion biases (Supplementary Fig. 2b, left panel). Similarly, spatial localization of the provirus and two representative RIGs remained unchanged upon treatment (Supplementary Fig. 2c, d).

Transcriptional profiling of activated CD4$^+$ T cells confirmed that protein-coding genes proximal to SEs are significantly more upregulated or downregulated upon JQ1 treatment than coding genes without SEs (Supplementary Fig. 2e, f). This effect is more pronounced among RIGs than among non-targeted genes (Supplementary Fig. 2f). Of note, HIV-1 maintains its preferences for highly transcribed genes in both control and JQ1-treated cells (compare Fig. 2a and Supplementary Fig. 2g).

In summary, our gene expression analysis suggests that genes recurrently targeted by HIV-1 are adjacent to SE elements, irrespective of their transcriptional levels, but disruption of SEs does not impact HIV-1 integration patterns.

**HIV-1 insertion hotspots are clustered in the nuclear space.** Our previously published results showed that the majority of tested RIGs are distributed in the outer zones of the T cell nucleus[18], so we hypothesized that the enrichment of HIV-1 insertion sites near SEs may be due to their particular organization in the nuclear space. We thus performed Hi-C to get some insight into the conformation of the T cell genome.

In order to minimize issues caused by the heterogeneity of the biological material, we used the widely available Jurkat lymphoid T cellular model. To ensure that the behavior of HIV-1 is similar in both models, we compared a published collection of 58,240 insertion sites in Jurkat cells[28] to the 28,419 insertion sites in primary CD4$^+$ T cells from the current study (obtained by linear amplification-mediated and inverse PCR) and previous

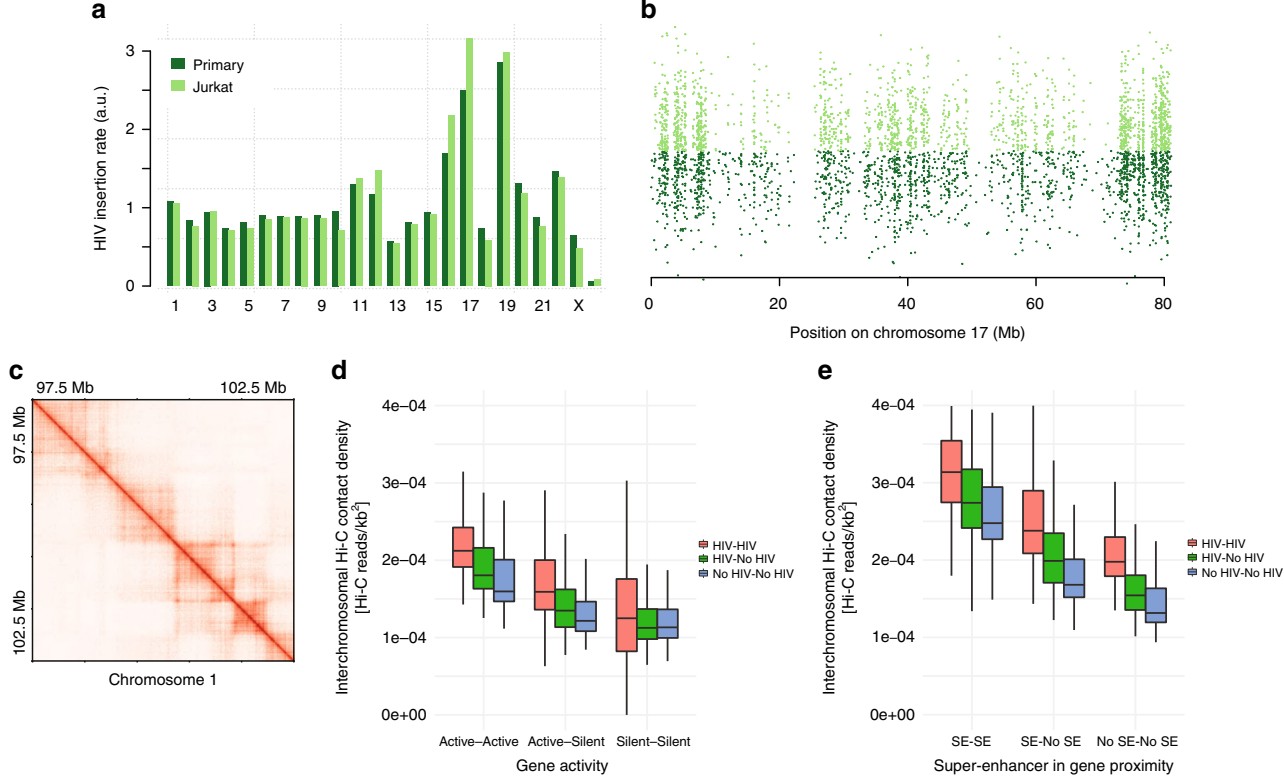

**Fig. 3** HIV-1 integration hotspots are clustered in the nuclear space. **a** Bar plot of HIV-1 insertion rate per chromosome (the genome-wide average is set to 1) in primary T and in Jurkat cells. **b** HIV-1 insertion cloud on chromosome 17 in primary T and Jurkat cells. Each dot represents an HIV-1 insertion site. The x-coordinate indicates the location of the insertion site on chromosome 17; the y-coordinate is random so that insertion hotspots appear as vertical lines. **c** Detail of the unnormalized Hi-C contact map in Jurkat in 5 kb bins. TADs and loop domains are clearly visible. **d** Box plot of inter-chromosomal Hi-C contact density (see "Methods"). Contact densities were computed between chromosomal aggregates of all gene fragments (5 kb) corresponding to Active and Silent genes, with (HIV) or without HIV insertions (No HIV). The distribution of densities are composed of the scores for all inter-chromosomal combinations. **e** Same as in **d**, but genes are classified between genes in proximity of super-enhancers (SE), i.e., within gene body or 5 kb upstream of TSS, or far from super-enhancers (No SE)

studies[38,39,51–54]. The insertion rates per chromosome are similar between cells (Fig. 3a); both show the characteristic approximately threefold increase on chromosomes 17 and 19. The apparent difference on chromosome 17 is possibly due to the use of different mapping technologies. For comparison, our previous measure of the insertion rates on chromosome 17 of Jurkat cells[29] (using the same inverse PCR technology) is very close to the current measure in primary CD4$^+$ T cells. The insertion cloud representation shows that the profiles are similar on chromosome 17, with the exception of a hotspot visible only in primary CD4$^+$ T cells at position ~57 Mb (Fig. 3b). We also found that the HIV-1 target genes are similar in Jurkat cells and in other CD4$^+$ datasets (Supplementary Fig. 3). In summary, apart from minor differences, HIV-1 insertion biases are comparable in primary CD4$^+$ T and in Jurkat cells.

Hi-C on uninfected Jurkat cells yielded ~1.5 billion informative contacts. Topologically associating domains (TADs) and loop domains are clearly visible on the raw Hi-C map in 5 kb bins (Fig. 3c), showing that the experiment captures the basic structural features of the Jurkat genome. We also verified that the A and B compartments are well defined and that they correspond to the regions of high and low gene expression, respectively (data not shown). To our knowledge, this dataset constitutes the highest-resolution Hi-C experiment presently available in Jurkat cells.

If the insertion pattern of HIV-1 reflects a particular organization of the genome, one predicts that the insertion hotspots occupy the same nuclear space and thus cluster together in three dimension (3D). We tested this hypothesis by measuring

the amount of inter-chromosomal Hi-C contact densities among different classes of HIV-1 insertion sites (Fig. 3d). The loci most targeted by HIV-1 engage in stronger contact with each other than non-targeted loci. Also, the differences in contact strength are more pronounced when loci correspond to active genes. In addition, SEs tend to cluster together and with HIV-1 insertion hotspots in 3D (Fig. 3e, Supplementary Fig. 4), indicating that SEs locate in the physical proximity of HIV-1 insertion sites. Thus HIV-1 insertion sites form spatial clusters interacting with SEs in the nucleus, consistently with the view that the insertion process depends on the underlying 3D organization of the T cell genome.

**SEs and HIV-1 occupy the same 3D sub-compartment.** To better define the properties of HIV-1 insertion sites, we segmented the Jurkat genome into spatial clusters. For each chromosome, we generated 15 clusters of loci enriched in self interactions, which we coalesced down to 5 genome-wide clusters based on their inter-chromosomal contacts (Fig. 4a and see "Methods"). This approach yielded two A-type sub-compartments called A1 and A2, two B-type sub-compartments called B1 and B2, and one intermediate/mixed compartment called AB (Fig. 4b, c).

The AB- and B-type sub-compartments correspond to known types of silent chromatin: AB is richest in the Polycomb mark H3K27me3, B1 is richest in H3K9me3, and B2 is richest in lamin (Fig. 4d and Supplementary Fig. 5a). The two A-type sub-compartments are enriched in euchromatin marks, with higher coverage in A1 than in A2 (Fig. 4d and Supplementary Fig. 5a).

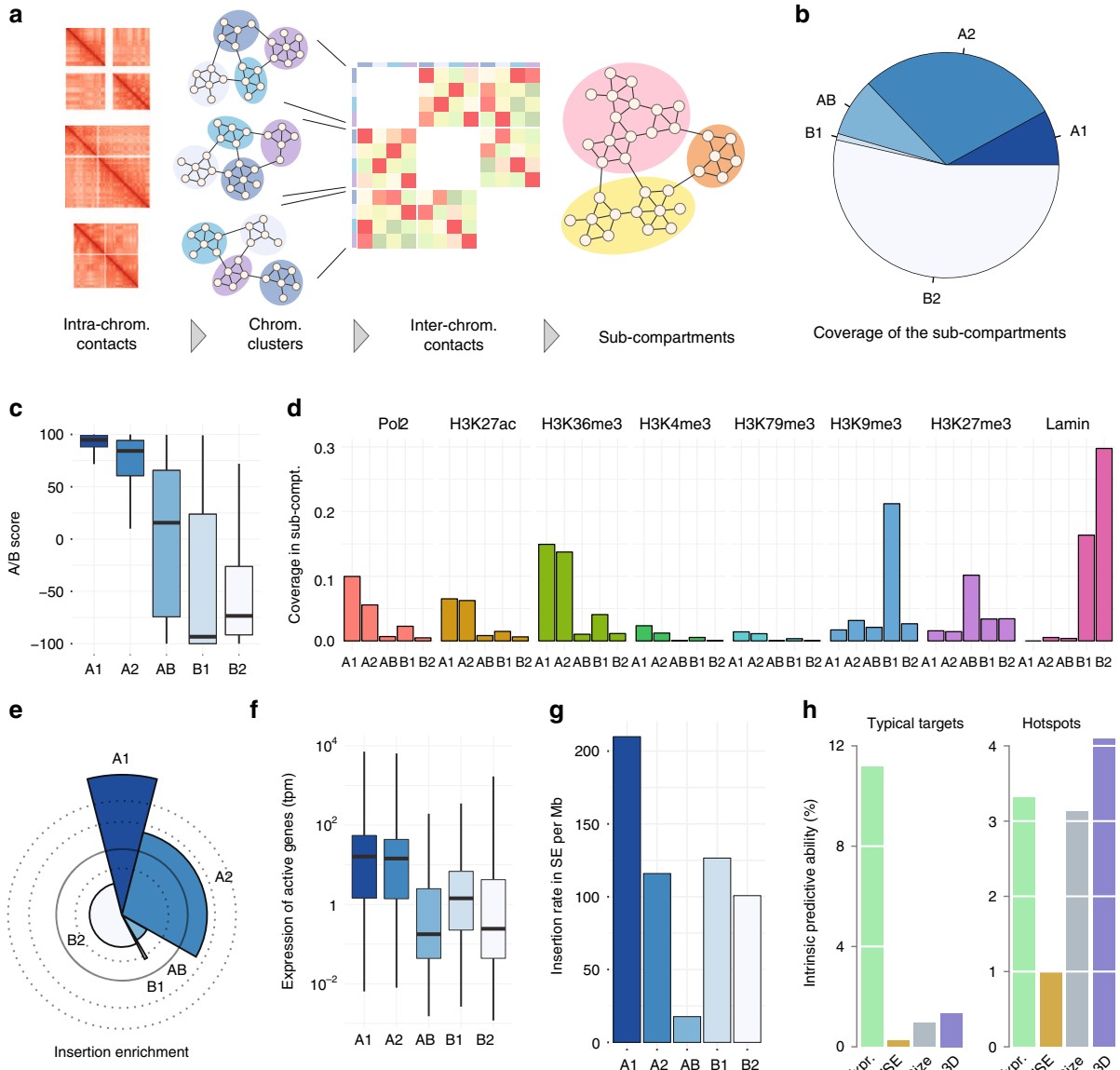

**Fig. 4** Super-enhancers and HIV-1 occupy the same 3D sub-compartment. **a** Definition and identification of 3D compartments. For each chromosome, 15 spatial communities were identified by clustering. The inter-chromosomal contacts between the communities were used as a basis for another round of clustering in five genome-wide spatial communities. **b** Pie chart showing the coverage of the sub-compartments in the Jurkat genome. **c** Distribution of AB scores in the 3D sub-compartments. The AB score measures the likelihood that a locus belongs to the A or B compartment. Extreme values +100 and −100 stands for "fully in A" or "fully in B", respectively. A score of 0 means "both or neither." **d** Proportion of 3D sub-compartments covered by major chromatin features. Coverage was computed as the span of enriched ChIP-Seq signal divided by the sub-compartment size. **e** Spie chart showing the observed vs expected HIV-1 insertions in the sub-compartments. The expected amount of insertions is the area of the wedge delimited by the circle in bold line, and the observed amount is the area of the colored wedge. Dotted lines represent the limit of the wedge for 2× and 3× enrichment outside the circle, and 0.5× depletion inside the circle. The observed/expected ratio is approximately 2.5 times higher in A1 than in A2. **f** Box plot showing the expression of protein-coding genes in the sub-compartments. The plot was rendered using defaults from the ggplot2 library. The y-axis has a logarithmic scale. **g** Bar plot showing the integration density of HIV in super-enhancers located in different 3D sub-compartments. **h** Bar plot showing the contribution of different predictors to the HIV-1 insertion sites in typical genes (left) or in hotspots (right). The y-axis represents the loss of accuracy when the corresponding variable is removed from the model. Expr. gene expression, dSE distance to nearest super-enhancer, Size gene size, 3D sub-compartment. See "Methods" for detail

Strikingly, the rate of HIV-1 insertion is 2.7 times higher in A1 than in A2 (Fig. 4e). In contrast, the coverage of euchromatin marks and the transcriptional activity are only slightly higher in A1 than in A2 (Fig. 4d, f), e.g., 1.04 times higher in H3K27ac coverage, 1.09 times in H3K36me3 coverage, and 1.12 times in median gene expression. More importantly, the ~2.5-fold enrichment of HIV-1 insertion is still present when controlling for gene expression (Supplementary Fig. 5b), indicating an intrinsic preference for the A1 sub-compartment. Of note, we

obtained similar results when defining 10 sub-compartments instead of 5, where HIV-1 insertion rates are enriched in one sub-compartment covering ~10% of the genome (data not shown). Hence, our observation is robust with respect to the definition of sub-compartments. The 3D organization of the Jurkat T cell genome thus explains large differences of HIV-1 insertion rates between genes expressed at similar levels.

If HIV-1 targets SEs because of their location in the nuclear space, one predicts that the insertion rate of HIV-1 in the SEs of

A1 should be higher than in the SEs of A2. Figure 4g shows that, indeed, HIV-1 is ~1.5 times more likely to integrate in the SEs of A1 than in those of A2. Since the insertion rate in SEs depends primarily on their location, we conclude that the enrichment in SEs at genome-wide scale is due to their position in the 3D space of the nucleus, rather than to their activity or their chromatin features.

To quantify this statement and to clarify how different determinants contribute to HIV-1 insertion, we used a modeling approach based on logistic regression. We predicted either typical HIV-1 target genes (top 33% gene-wide insertion rate) or HIV-1 hotspots (top 2.5% bin-wise insertion rate, see "Methods"). Typical HIV-1 targets are almost entirely determined by gene expression (Fig. 4h), consistently with previous reports that HIV-1 integrates primarily in active genes[2,3,20]. On the other hand, HIV-1 hotspots are multi-factorial and sub-compartments appear as the major determinants (Fig. 4h). These results establish that typical HIV-1 targets and hotspots, such as RIGs, are driven by different classes of mechanisms. Finally, they show that the 3D organization is a major contributor of HIV-1 hotspots.

**Genes proximal to SEs reposition upon T cell activation**. Our results so far suggest that HIV-1 insertion hotspots cluster near SEs because of their location in the structured genome of T cells, but they do not address the contribution of SEs to this structure.

We thus investigated the role of SEs in the spatial distribution of genes in T cells. RIGs belong to a subset of T cell genes that show the strongest response to T cell activation (Supplementary Fig. 6a), so we reasoned that their spatial positioning might change with the activation status of the cell. We therefore employed 3D immuno-DNA fluorescence in situ hybridization (FISH) to visualize gene positioning in resting and activated CD4+ T cells. The cumulative frequency plots revealed that nine RIGs, seven of which have SEs FOXP1, STAT5B, NFATC3 (Fig. 5a), KDM2A, PACS1 (Fig. 5b), and GRB2, RNF157 (Fig. 5c), change spatial positioning and relocalize further toward the outer shells of the T cell nucleus upon activation. Three RIGs, NPLOC4, RPTOR, and BACH2, were already peripheral before activation and remained so afterwards (Supplementary Fig. 6c). We recapitulated the overall distribution of 9 RIGs that displayed repositioning in activated ($n = 1690$ alleles) vs resting CD4+ T cells ($n = 1700$ alleles, Fig. 5d). As expected, the frequency distribution of alleles in three zones of equal surface areas[18] showed a prominent shift toward the outer shells of the nucleus in activated T cells, corresponding to the area located <1 micron under the nuclear envelope. Of note, a pan nuclear distribution of KDM2A and PACS1 in activated CD4+ T cells was also observed[26].

We then asked whether the observed gene redistribution is an exclusive feature of genes proximal to SEs or a general feature of all expressed genes, independent of HIV-1 targeting. We therefore evaluated the spatial distribution of two groups of control genes: expressed genes with SEs and expressed genes without SEs. The MYC gene, a gene harboring five well-described SEs, changed its radial position toward the outer shells of the nucleus upon T cell activation (Supplementary Fig. 6d). The same trend was observed for two other regions proximal to SEs that are not targeted by the virus: one on chromosome 1 covering the gene LMNA and the other on chromosome 11 encompassing SLC43A1, UBE2L6, and TIMM10. Both regions showed statistically significant repositioning toward the more exterior shells of the nucleus with T cell activation.

In contrast, when we assessed the spatial distribution of three highly expressed genes without SEs, TAP1, CCNC, and MCM4,

we did not observe any statistically significant allele redistribution upon T cell activation (Supplementary Fig. 6e).

Next, we wanted to understand whether disruption of SEs impacts the nuclear position of genes proximal to these elements during T cell activation. To do so, we pretreated resting T cells with JQ1 before activating them with CD3/CD28 beads and observed that the two tested RIGs, STAT5B and GRB2, retained their position in the center of the nucleus (Supplementary Fig. 6g, h), supporting the notion that SEs contribute to the positioning of genes prior and during T cell activation.

As HIV-1 target genes group together on linear chromosomes[18] and integration hotspots cluster in the nuclear space (Fig. 3e and Supplementary Fig. 4), we assessed their spatial relationships during T cell activation. Two highly targeted regions (top 10% of RIGs density) on chromosomes 11 and 17 were visualized by dual-color FISH coupled to high-throughput imaging (HTI)[65,66]. We observed that KDM2A and PACS1, two genes proximal to SEs lying at 1.1 Mb from each other on chromosome 11 (Fig. 5e), clustered together in the nuclear space, with a minimized median distance of 0.42 μm in both resting and activated state (data summarized in Supplementary Fig. 6f). Similarly, we found that, in the hotspot region mapping to q25.1-3 on chromosome 17 containing 35 RIGs (Fig. 5f), three RIGs, GRB2, TNRC6C, and RNF157, cluster together (Fig. 5f, data summarized in Supplementary Fig. 6f). This clustering is not a mere consequence of the linear distances between these genes, as two other genes from the same locus, NPLOC4 and RPTOR, despite being at a similar linear distance, are not spatially associated (measured distances given in Supplementary Fig. 6f).

In summary, our results show that seven out of nine RIGs proximal to SEs change their radial positioning upon T cell activation, moving to the outer shell of the nucleus. This is a feature pertinent also to genes proximal to SEs that are not HIV-1 targets, suggesting that SEs contribute to the spatial organization of the genome and that in dependence of the activation state could be more exposed to HIV-1 insertions.

## Discussion

The integration of the viral DNA into the host cell genome is responsible for the long-term persistence of HIV-1 in cellular reservoirs[67]. The persistence of HIV-1 is influenced by the chromosomal context at the sites of integration, with a strong impact on the outcome of viral infection[29]. Here we characterized the genomic features of integration sites identified from patients[38,39,51–54] and from in vitro infections of activated CD4+ T cells (ref. [50] and this study). By analyzing these large datasets, we confirmed that HIV-1 recurrently integrates into a subset of transcriptionally active cellular genes. We show that HIV-1 recurrently integrates into a group of genes proximal to SE genomic elements in activated CD4+ T cells and in patients (Fig. 1c). Yet, neither the activity of SEs nor their effect on gene expression alone explain the integration biases (Supplementary Fig. 2b). Instead, we found that the correlation can be attributed to the enrichment of SEs in the A2 and especially in the A1 sub-compartments, where HIV-1 integrates at higher frequency than in the rest of the genome (Fig. 4e).

The contribution of gene expression levels to the insertion rate of HIV-1 is intricate. On one hand, HIV-1 shows a clear bias toward expressed genes, even upon JQ1 treatment where it still integrates preferentially into genes that are most active after JQ1 treatment. However, HIV-1 recurrently integrates into genes proximal to SEs (Fig. 2d), among which there are both upregulated and downregulated genes (Supplementary Fig. 2f). One potential explanation is that the HIV-1 IN has a strong affinity for some protein present in transcribed regions (e.g., LEDGF). The

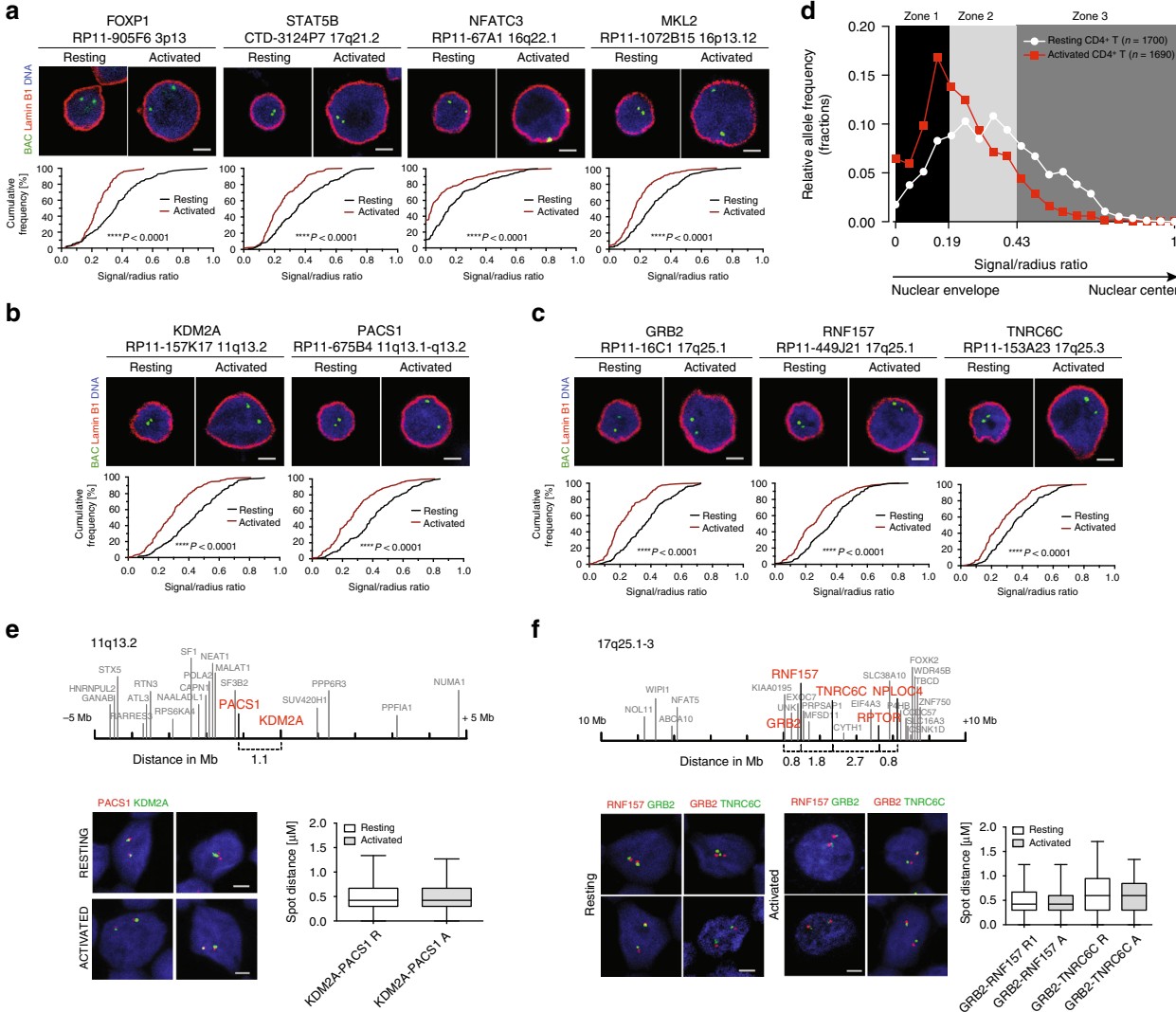

**Fig. 5** Genes proximal to super-enhancers change their nuclear positioning upon T cell activation. Three-dimensional immuno-DNA FISH of nine RIGs in resting and activated (anti-CD3/anti-CD28 beads, IL-2 for 48 h) CD4$^+$ T cells (green: BAC/gene probe, red: lamin B1, blue: DNA staining with Hoechst 33342, scale bar represents 2 µm). Cumulative frequency plots show combined data from both experiments ($n = 100$, black: resting cells, red: activated cells). The $p$ values of the Kolmogorov–Smirnov tests are indicated. Box plots represent minimized distances (5th−95th percentile) for the analyzed gene combinations in resting (white) and activated (gray) CD4$^+$ T cells, obtained by high-throughput imaging and subsequent computational measurements. In the box plots, the center line represents the median, the bounds of the box span from 25% to 75% percentile, and the whiskers visualize 5% and 95% of the data points. Representative images for **a** *FOXP1*, *STAT5B*, *NFATC3*, and *MKL2*; **b** *KDM2A* and *PACS1*; and **c** *GRB2*, *RNF157*, and *TNRC6C*. **d** Allele fraction density plot for all resting and activated alleles that displayed peripheral repositioning. The *y*-axis shows the allele fraction density for genes *FOXP1*, *STAT5B*, *NFATC3*, *MKL2*, *KDM2A*, *PACS*, *GRB2*, *RNF157*, and *TNRC6C*. The *x*-axis represents ratios of distance from nuclear envelope (lamin B1 staining) and radius (signal to radius ratio) for alleles in resting cells ($n = 1700$) and activated cells ($n = 1690$ alleles). Binning into three equal concentric zones of the nucleus is performed as in ref. [18]. **e** Schematic representation of chromosomal region 11q13.2 within 10 Mb: RIGs (bold red) and single HIV-1 integration sites (plain gray) and HTI of *KDM2A* and *PACS1*. **f** Schematic representation of chromosomal region 17q25.1-3 within 20 Mb: RIGs (bold red) and single HIV-1 integration sites (plain gray) and HTI of *GRB2*, *RNF157*, and *TNRC6C*

complete absence of such proteins in non-transcribed regions would have more influence on the signal than its quantitative variations in transcribed regions. In any event, gene expression and chromatin are not the sole contributors to HIV-1 insertion patterns. The A1 sub-compartment is targeted more frequently than the rest of the genome (Fig. 4e), even when controlling for chromatin and gene expression (Supplementary Fig. 5b). This indicates that the 3D genome organization of activated T lymphocytes is an important determinant of the HIV-1 insertion process. SEs most likely contribute to this organization[42] because their dismantling prior to T cell activation prevents repositioning

of genes with SEs toward the outer shells of the nucleus (Supplementary Fig. 6g, h).

Predictive modeling helps clarify this conclusion. An important insight is that HIV-1 insertion hotspots do not obey the same rules as typical target genes (Fig. 4h). There are thus two processes at work: one that attracts viruses to active genes, and another one, more complex, that provokes recurrent integrations within the same genes (i.e., the RIGs). 3D compartmentalization plays a role only in the second process, explaining why studies with different definitions of HIV-1 targets may come to different conclusions.

Although we show that SEs do not affect HIV-1 integration patterns in activated T cells, we find that, during T cell activation, genes with SEs move toward the outer shells of the nucleus (Fig. 5a–c). In line with the previously shown association of nuclear pore proteins with HIV-1[18,19], and their proximity to SEs and enrichment in the A1 sub-compartment defined here (ref. [41] and data not shown), it is tempting to speculate that the A1 sub-compartment corresponds to genomic loci associated with the nuclear pore. None of the chromatin features mapped in Jurkat cells is known to discriminate active genes at the nuclear pore from other active genes, and the chromatin of A1 is otherwise similar to that of A2 (Supplementary Fig. 5a). Interestingly, the density of SEs is similar between A1 and A2 (Supplementary Fig. 5a), so it is unlikely that the A1 sub-compartment simply emerges from the clustering of SEs. More plausibly, SEs are one of many contributors to the segregation of the genome in spatial clusters. More generally, the existence of two separate clusters of active genes in the 3D space of the nucleus is itself an intriguing observation that will require more work to be fully understood.

While the spatial positioning of the A1 and A2 sub-compartments in T cells still needs to be mapped, a recent study proposed an alternative concept to the one where nuclear periphery represents solely transcriptionally repressive environment[68,69]. Instead, and consistently with our findings, they predict distinctive localization of active A1 and A2 Hi-C sub-compartments. Transcriptionally active regions are divided into two groups: a transcriptional "hot zone" close to nuclear speckles corresponds to the A1 sub-compartment and another one far from speckles corresponds to A2[68]. Interestingly, transcriptional hot zones confined within the A1 sub-compartment are enriched in SEs and highly expressed genes, traits we observed to be strongly associated with HIV-1 insertional hotspots.

It is well established that the main components that mediate HIV-1 integration into actively transcribing units are the viral proteins IN and CA[3,17]. Their cellular partners LEDGF/p75 and CPSF6 could chaperone the virus into clusters of SE domains in the A1 compartment. LEDGF/p75 interacts with a large number of splicing factors and directs HIV-1 integration to highly spliced transcription units[22], making this a plausible link to the A1 compartment.

Likewise, CPSF6 as part of the mRNA polyadenylation machinery, could guide HIV-1 integrations toward the nuclear compartment with high transcription and mRNA processing rates (such as A1[68]). Alternatively, the CA–CPSF6 axis could regulate HIV-1 targeting independently of the polyadenylation role of CPSF6[24,26,70].

Among the factors that are binding putative SEs and could play a role in integration site selection, p300 and BRD4 seem to be the most promising candidates. p300, a histone acetyltransferase used to identify typical[71,72] and SEs[32,73,74], is an interaction partner of the HIV-1 IN. p300 promotes the DNA-binding activity of IN[75] and could serve to direct viral integration toward genes with SEs in the A1 compartment, though a role for p300 in HIV-1 integration targeting has yet to be established.

BRD4, on the other hand, does not bind HIV-1 IN[76,77] but has a well-established role in HIV-1 latency[78,79]. The mechanism of action has recently been ascribed to the short isoform of BRD4, which recruits a repressive SWI/SNF complex to the viral long terminal repeat (LTR)[80]. Loss of the short isoform, occurring rapidly upon JQ1 treatment, leaves the long isoform engaged in the transcriptional activation of the viral genome[80]. The same mechanism could account for the activation of cellular genes upon JQ1 treatment[31]. In fact, our RNA-Seq data show that genes proximal to SEs are both upregulated and downregulated upon JQ1 (Supplementary Fig. 2e). Furthermore, genes targeted by HIV-1 are more responsive to JQ1 than non-HIV-1 targets

(Supplementary Fig. 2g). This implies that HIV-1 preferentially targets genes that have a rapid and tightly regulated transcriptional response. Given the opposing role of BRD4 on viral LTR and cellular genes, insertion into genes proximal to SEs might represent a source of transcriptional fluctuations[81,82] and play an important role in either establishment or reversal of latency.

Based on our findings that the majority of tested RIGs and genes with SEs reposition from the nuclear interior to the periphery during T cell activation, it could be envisaged that RIGs differ between resting and activated CD4[+] T cells. Meta-analysis of the only available integration sites dataset[50] from these two cell activation states showed, however, no significant difference. Additional work will thus be required to assess comprehensively the RIGs that are used by HIV-1 in resting T cells.

Overall, we show that HIV-1 insertion sites form spatial clusters interacting with SEs of A1 compartment, highlighting the importance of the underlying 3D genome organization for HIV-1 integration. While additional studies will be needed to decipher the mechanism of such site selection, our results identify hotspots of integration that could improve characterization and enable targeting of latent HIV-1 reservoirs.

## Methods

**Primary cell isolation, culture, treatments, and infection.** For CD4[+] T cells isolation, whole blood was mixed with RosetteSep Human CD4[+] T cell enrichment cocktail beads according to the manufacturer's instructions and CD4[+] T cells were separated using Histopaque Ficoll gradient by centrifugation. Cells were cultured in complete T cell medium (RPMI-1640+10% fetal bovine serum (FBS) + primocin), left in resting state or activated with Dynabeads Human T-Activator CD3/CD28 and plated in complete medium supplemented with 5 ng/ml IL-2 for 20–72 h at 37 °C.

Cells were treated when indicated with 500 nM JQ1(+) or dimethyl sulfoxide (DMSO) for 6 h at 37 °C.

In all, $1 \times 10^6$ activated CD4[+] T cells were infected with 0.5–1 μg of p24 of virus by spinoculation for 90 min at 2300 rpm at room temperature (RT) in the presence of polybrene at 37 °C. Virus stocks were produced from the viral clone HIV-1NL$_{4_3}$ and a mutant that harbors a frameshift (FS) mutation in the *env* gene (pNL$_{4_3}$-envFS) and was pseudotyped with vesicular stomatitis virus glycoprotein, resulting in a FS virus that performs a single-round infection (HIV-1NL$_{4_3}$ FS). Cells were then incubated for 72 h at 37 °C. When indicated, 14 h after infection with HIV-1NL$_{4_3}$, cells were treated with the fusion inhibitor T20 to prevent multiple infection and integration. All viral stocks were generated by transfecting viral DNA in HEK 293T cells and collecting supernatants after 48–72 h following sucrose gradient purification of virus articles. Viral production was quantified in the supernatants for HIV-1 p24 antigen content using the Innotest HIV Antigen mAB Kit (INNOGENETICS N.V. Gent, Belgium). The human Jurkat T cell line (obtained from the cell collection of the Center for Genomic Regulation, Barcelona) was grown at 37 °C under a 95% air and 5% $CO_2$ atmosphere, in RPMI 1640 medium (Gibco) supplemented with 10% FBS (Gibco), 1% penicillin–streptomycin (Gibco), and 1% GlutaMAX (100×) (Gibco). Jurkat cells were passaged every 2 days with a 1:5 dilution. Cells were tested for mycoplasma regularly.

**Fluorescence in situ hybridization.** Approximately $3 \times 10^5$ CD4[+] T cells were plated on the PEI-coated coverslips placed into a 24-well plate for 1 h at 37 °C. Cells were treated with 0.3× phosphate-buffered saline (PBS) to induce a hypotonic shock and fixed in 4% paraformaldehyde (PFA)/PBS for 10 min Coverslips were extensively washed with PBS and cells were permeabilized in 0.5% triton X-100/PBS for 10 min. After three additional washings with PBS-T (0.1% tween-20), coverslips were blocked with 4% bovine serum albumin (BSA)/PBS for 45 min at RT and primary antibody anti-lamin B1 ab16048, from Abcam (1:500 in 1% BSA/PBS), was incubated overnight at 4 °C. Following three washings with PBS-T, fluorophore-coupled secondary antibody (anti-rabbit, coupled to Alexa 488 #11034, Alexa 568 #A11011, or Alexa 647 #A27040 from Invitrogen, diluted 1:1000 in 1% BSA/PBS) were incubated for 1 h at RT, extensively washed, and post fixed with ethylene glycol bis(succinimidyl succinate) (EGS) in PBS. Coverslips were washed three times with PBS-T and incubated in 0.5% triton X-100/0.5% saponin/PBS for 10 min. After three washings with PBS-T, coverslips were treated with 0.1 M HCl for 10 min, washed three times with PBS-T, and additionally permeabilized step in 0.5% triton X-100/0.5% saponin/PBS for 10 min. After extensive PBS-T washings, coverslips were equilibrated for 5 min in 2× saline sodium citrate (SSC) and then put in hybridization solution overnight at 4 °C. For the HIV-1 FISH, RNA digestion was additionally performed beforehand using RNAse A (100 μg/ml).

For FISH without immunofluorescence (IF) for HTI, $1–2 \times 10^6$ CD4[+] T cells in 500 μl of medium were adhered to coverslips by centrifugation at $350 \times g$ for 10

min at RT. The coverslips were washed in PBS and the cells were fixed in 4% PFA/PBS for 10 min followed by extensive PBS washing. Permeabilization was performed by incubation in 0.5% triton X-100/0.5% saponin/PBS for 20 min. After three washings with PBS, cells were treated with 0.1 M HCl for 15 min. Coverslips were washed twice for 10 min with 2× SSC and put in hybridization solution overnight at 4 °C.

For DNA probe labeling, bacterial artificial chromosome (BAC) or P1 artificial chromosome (PAC) DNA was extracted using a Nucleobond Xtra Maxiprep or amplified by the Illustra GenomiPhi V2 DNA Amplification Kit according to the manufacturer's instructions. HIV-1 plasmid HXB2 was purified using the Qiagen Plasmid Extraction Kit. FISH probes were generated in a Nick translation reaction using three different protocols. All BACs/PACs are listed in Supplementary Table 5.

BACs were labeled with digoxigenin (DIG)-coupled dUTPs. Three micrograms of BAC DNA were diluted in $H_2O$ in a final volume of 16 µl. Four microliters of DIG-Nick translation mix (Roche) were added and the labeling reaction was carried out at 15 °C for up to 15 h. The labeling reaction was performed by using a fluorophore-coupled dUTPs in the same concentration as biotin-16-dUTP in ref. [2].

For HIV-1 labeling, a biotin-dUTP nucleotide mix containing 0.25 mM dATP, 0.25 mM dCTP, 0.25 mM dGTP, 0.17 mM dTTP, and 0.08 mM biotin-16-dUTP in $H_2O$ was prepared. Three micrograms of pHXB2 were diluted with $H_2O$ in a final volume of 12 µl, and 4 µl of each nucleotide mix and Nick translation mix (Roche) were added. Labeling was performed at 15 °C for 3–6 h.

For dual-color FISH or improvement of signal-to-noise ratio in single-color FISH, probes were labeled using the fluorophore-coupled nucleotides SpectrumGreen dUTP (Abbott), SpectrumOrange dUTP (Abbott), and Red 650 dUTP (Enzo).

In all, 1–3 µg of BAC DNA were diluted in a final volume of 22.5 µl $H_2O$. Also, 2.5 µl of 0.2 mM fluorophore-coupled dUTP, 5 µl of 0.1 mM dTTP, 10 µl of dNTP mix containing 0.1 mM of each dATP, dCTP, and dGTP, and 5 µl of 5× Nick translation buffer (Abbott) were added and reagents were mixed well by vortexing. The reaction was started by addition of 5 µl Nick translation enzymes (Abbott) and incubated at 15 °C for 13–14 h. The probes were checked for their size on a 1% agarose gel, and 200–500 bp probes were purified using Illustra Microspin G-25 columns according to the manufacturer's instructions. Probes were precipitated in ethanol, dissolved in formamide and 4× SSC/20% dextran sulfate (1:1), and stored at −20 °C prior to use.

For probe hybridization, 1–6 µl of probe was loaded on glass coverslips and heat denatured in metal chamber at 80 °C for 8 min in a water bath. Hybridization was carried out for 48 h at 37 °C. Four washings in 2× SSC (10 min each) at 37 °C were followed with 2 washings in 0.5× SSC at 56 °C.

FISH development for DIG-labeled BACs was performed by using fluorescein isothiocyanate (FITC)-labeled anti-DIG antibody (Roche), whereas biotin-labeled HIV-1 probes were detected by TSA Plus system from Perkin Elmer, that allows significant amplification of the signal, by using an anti-biotin antibody (SA-HRP) and a secondary antibody with a fluorescent dye (usually FITC for HIV).

For the directly labeled probes after initial washings, nuclei were stained with Hoechst 33342 (1:5000 in PBS), washed in PBS, and then mounted using mowiol.

**Microscopy and image analysis**. For the classical confocal microscopy and manual image analysis, 3D stacks were acquired with a Leica TCS SP8 confocal microscope using a ×63 oil immersion objective. Distance measurements were performed using Volocity (Perkin Elmer). The smallest distance between the FISH signal and the nuclear lamina, stained by IF for lamin B1, was determined, and measurements were normalized to the nuclear radius (defined as half of the maximum diameter of the lamin B1 ring). Signal-to-radius ratios were either binned into three classes of equal surface (zones 1–3)[18] or plotted on a cumulative frequency plot. Kolmogorov–Smirnov (KS) tests were performed to compare the distributions of positioning of a gene between two conditions (resting vs activated or DMSO vs JQ1).

For HTI and image analysis of dual-color FISH, images were acquired with a spinning disk Opera Phenix High Content Screening System (PerkinElmer), equipped with four laser lines (405 nm, 488 nm, 568 nm, 640 nm). Images of FISH experiments to calculate 3D distances were acquired in confocal mode using a ×40 water objective lens (NA 1.1) and two 16 bit CMOS cameras (2160 by 2160 pixels), with camera pixel binning of 2 (corresponding to 299 nm pixel size). For each sample, 11 z-planes separated by 0.5 µm were obtained for a total number of at least 36 randomly sampled fields, which acquired per condition a minimum of $16 \times 10^3$ cells. Image analysis was performed using the Harmony high-content imaging and analysis software (version 4.4, Perkin Elmer), using custom-made image analysis building blocks. Nuclei were segmented based on the Hoechst nuclei staining signal of maximum projected images using the algorithm B and cells in the periphery of the image were excluded from further analysis. FISH probe detection was performed by using the spot detection algorithm C and custom-made scripts were used to calculate the Euclidean distances between all the different colored probes per cell. Single cell-level data were then exported and custom-made R scripts were used to select the minimum distance between the different FISH probes per allele basis. To exclude spurious spot detection events from the analysis, only the distances of cells with two FISH probes detected per channel were calculated and plotted (Graph Pad, Prism).

**Quantitative real-time PCR (qPCR)**. Up to $5 \times 10^6$ CD4$^+$ T cells were used for RNA extraction with the InviTrap Spin Kit (Stratec Biomedical) according to the manufacturer's instructions and up to 500 ng of RNA was retro-transcribed using Moloney MLV reverse transcriptase from Invitrogen according to the manufacturer's instructions. Gene expression analysis were performed in duplicates using IQ supermix from Biorad in CFX96/C1000 Touch Real-Time PCR system, as described in Lusic et al., 2013. Statistical analysis of qPCR data was performed using Graphpad. Taqman assays used were: for MYC Taqman Hs00153408_m1 FAM/MGB and for GAPDH 4310884E VIC/TAMRA.

**Western blotting**. In all, $5 \times 10^6$ cells were harvested and homogenized in lysis buffer (20 mM Tris-HCl, pH 7.4, 1 mM EDTA, 150 mM NaCl, 0.5% Nonidet P-40, 0.1% sodium dodecyl sulfate (SDS), 0.5% sodium deoxycholate) supplemented with protease inhibitors (Roche) for 10 min at 4 °C and sonicated (Bioruptor) for 5 min. Equal amounts of total cellular proteins (20 µg), as measured with Bradford reagent (Biorad), were resolved by 10% SDS-polyacrylamide gel electrophoresis, transferred onto nitrocellulose membrane (GE Healthcare), and then probed with primary antibody, followed by secondary antibody conjugated with horseradish peroxidase. The immuno-complexes were visualized with enhanced chemiluminescence kits (GE Healthcare). Antibodies used were: for MYC 9E10, # sc-40 (1:500) from Santa Cruz and for actin Anti-β-Actin AC-74, # A5316 (1:5000) from Sigma Aldrich.

**Flow cytometric analysis**. T cell activation with CD3/CD28 activating beads was controlled with CD25 and CD69 activation markers. Approximately 150,000 were fixed in 3% PFA for 10 min at RT. Cells were washed in 1% FBS/PBS and stained with the corresponding antibody for 45 min on ice, (1:50 dilution was used for CD25 FITC, #555431 from BD and CD69 BV510 #310929 from Biolegend). Cells were extensively washed and profiled using BD FACSVerse™ instrument. Gates for activation marker-positive cells were set by utilizing unstained controls. FlowJo software was used for the data analysis. Gating strategy is described in Supplementary Fig. 7.

**Chromatin immunoprecipitation**. In all, $20 \times 10^6$ CD4$^+$ T cells were washed 1 time in PBS prior to crosslinking with 1% formaldehyde for 10 min at RT, followed by termination of the reaction with 125 mM glycine on ice. Cell pellet was washed 2 times with PBS at 4 °C and was lysed in 0.5% NP-40 buffer (10 mM Tris-Cl pH 7.4, 10 mM NaCl, 3 mM MgCl$_2$, 1 mM PMSF, and Protease Inhibitors). For histone ChIPs, obtained nuclei were washed once in the same buffer without NP-40. Nuclei were resuspended in 0.5% NP-40 buffer supplemented with 0.15% SDS and 1.5 mM CaCl$_2$. Nuclei were incubated at 37 °C for 10 min prior to addition of Micrococcal Nuclease (16 units of the enzyme), and the reaction was stopped after 7 min with 3 mM EGTA. DNA was additionally sheared by sonication (Covaris or Bioruptor, Diagenode) to an average size of DNA fragments <500 bps. Extracts were then diluted up to 0.01% SDS, 1% Triton-X, 20 mM Tris pH 8, 150 mM NaCl, and 2 mM EDTA. Extracts were precleared by 1-h incubation with protein A/G Magna ChIP beads at 4 °C and diluted with 5× IP buffer to a final concentration of 140 mM NaCl and 1% NP-40. Lysate corresponding to $3-4 \times 10^6$ million of cells was then incubated with 2–4 µg of the indicated antibody overnight at 4 °C, followed by a 2.5-h incubation with Magna ChIP Protein A/G Magnetic Beads (Millipore). Beads were then washed thoroughly with RIPA150, with LiCl-containing buffer and with TE buffer, RNAse treated for 1 h at 37 °C, and Proteinase K treated for 2 h at 56 °C. Decrosslinking of protein–DNA complexes was performed by an overnight incubation at 65 °C. Additional 1 h of Proteinase K digestion was performed at 56 °C and DNA was then extracted using Agencourt AMPure XP beads (Beckman Coulter) and quantified by real-time PCR. The following antibodies were used for ChIP: H3K27ac (ab4729), H3K4me3 (ab8580) H3K36me3 (ab9050), IgG Rabbit (ab46540).

**ChIP-Seq and RNA-Seq**. ChIP-Seq: Approximately 10 ng of the corresponding inputs and ChIP-ed DNA from primary CD4$^+$ T cells: H3K27ac, H3K4me3, H3K36me3, H4K20me1, and H3K9me2, IPs were prepared for sequencing using the NEBNext® Ultra™ II DNA Library Prep Kit for Illumina®.

RNA-Seq: $5 \times 10^6$ DMSO and 500 nM JQ1-treated CD4$^+$ T cells from three independent donors were used for RNA extraction with the InviTrap Spin Kit (Stratec Biomedical) according to the manufacturer's instructions and libraries for sequencing were prepared by using the rRNA Depletion Kit NEBNext® and NEBNext® Ultra™ RNA Library Prep Kit for Illumina®. Sequencing was performed with $2 \times 75$ bp read length on the NextSeq platform.

**In situ Hi-C protocol**. Hi-C was performed based on the protocol published by Rao et al.[83] with modifications. Briefly, one million cells were crosslinked with 1% formaldehyde for 10 min at RT with gentle rotation. Nuclei were permeabilized by 0.25 ml freshly prepared ice-cold Hi-C lysis buffer [10 mM Tris-HCl pH 8.0, 10 mM NaCl, 0.2% Igepal CA630 (Sigma, I8896–50ML), and 1× Roche complete protease inhibitors (Roche, 11836153001)]. DNA was digested with 100 units of MboI (NEB, R0147M) at 37 °C overnight, and the ends of digested fragments were filled in by using 0.4 mM biotinylated deoxyadenosine triphosphate (biotin-14-dATP; Life Technologies, #65001) and ligated in 1 ml by incubating at 24 °C

overnight with gentle rotation. After reversal of the crosslinks, ligated DNA was purified and sheared to a length of 400 bp. Ligation junctions were pulled down with 75 μl of 10 mg/ml streptavidin C1 beads. Ten microliters of DNA-on-beads were amplified in 50 μl standard Herculase II Fusion DNA Polymerase reaction mix (Agilent Technologies, #600675) with 1 μM NEBNext Universal primer and index primer (NEB, E6040S). The cycling conditions were as follows: 98 °C for 2 min; 98 °C for 20 s, 65 °C for 30 s, and 72 °C for 45 s (8 cycles); and 72 °C for 3 min. PCR products were purified with 1.0× Agencourt AMPure XP beads (BECKMAN COULTER, A63880). Libraries ran as a smear on 1.5% agarose gel and estimate of the size of a smear was around 300 bp. The quality of the libraries was assessed by digesting with ClaI (NEB, R0197S) and checking that the smear shifts downwards.

**Reporting summary**. Further information on research design is available in the Nature Research Reporting Summary linked to this article.

## Data availability

All relevant data supporting the key findings of this study are available within the article and its Supplementary Information files or from the corresponding authors on reasonable request. The RNA- Seq of resting and activated CD4+ T cells is available from Gene Expression Omnibus (GEO; Lucic et al. GSE122735), ChIP-Seq on primary CD4+ T cells (Lucic et al. GSE GSE122826), in situ Hi-C data (Chen et al. GSE122958). Integration site raw data on in vitro infected CD4+ T cells are available from GSE134382. A reporting summary for this Article is available as a Supplementary Information file.

## Code availability

Code for processing raw sequences to get integration sites is available here: https://github.com/gui11aume/genome_structure_and_HIV_integration/blob/master/maja/Brady_Integration_Sites.md. Integration sites used in this analysis are available as an R object containing a list of GRanges objects, one list element for each dataset used; https://github.com/gui11aume/genome_structure_and_HIV_integration/blob/master/maja/is.Robj. The matrix that contains the number of lists each gene is found in all 100 randomizations can be found here (in RDS format): https://github.com/gui11aume/genome_structure_and_HIV_integration/raw/master/maja/Replicates.RDS. Genuine Hi-C contacts were validated with the Hi.C pipeline (https://github.com/ezorita/hi.c).

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

## Acknowledgements

We thank the Infectious Diseases Imaging Platform (IDIP) and the platform coordinator Dr. Vibor Laketa (DZIF), as well as the genomics core facility of the CRG for their technical support. We also thank Monsef Benkirane and Thomas Gayraud from the IGH Montpellier for providing samples of cell sorted infected primary CD4+ T cell. This work was supported by German Center for Infection Research (DZIF) Thematic Translational Unit HIV-1 04.704 Infrastructural Measure to M.L. and by the Hector Grant M70 "HiPNose: HiV Positioning in the Nuclear Space" to M.L. and M.S. We acknowledge the financial support of the Spanish Ministry of Economy and Competitiveness ("Centro de Excelencia Severo Ochoa 2013–2017," Plan Nacional BFU2012–37168), of the CERCA (Centres de Recerca de Catalunya) Programme/Generalitat de Catalunya, and of the European Research Council (Synergy Grant 609989). K.V. and M.K. are supported by the European Structural and Investment Funds grant for the Croatian National Centre of Research Excellence in Personalized Healthcare (contract #KK.01.1.1.01.0010), Croatian National Centre of Research Excellence for Data Science and Advanced Cooperative Systems (contract KK.01.1.1.01.0009), and Croatian Science Foundation (grant IP-2014–09–6400). This work was funded in part by the Deutsche Forschungsgemeinschaft (DFG, German Research Foundation) – Project number 240245660- SFB 1129, Project 20.

## Author contributions

M.L, B.L. and G.F. designed the research; B.L., H.C., J.W., V.M. and W.W. performed the experiments; M.K., R.S., E.Z., R.F., K.V., and G.F. performed bioinformatics analysis; B.L., M.K., E.Z., V.R., K.V., G.F. and M.L analyzed the data; B.L., M.K., G.F. and M.L. wrote the manuscript; M.L., G.F., K.V., M.S. and S.L. provided funding.

## Additional information

**Competing interests:** The authors declare no competing interests.

