## [Peer Review File · Nature Communications]

Reviewers' comments:

Reviewer #1 (Remarks to the Author):

Spatially clustered loci with multiple enhancers are frequent targets of HIV-1, Lucic et. al

This manuscript describes an analysis of HIV-1 integration patterns in the context of three-dimensional chromatin organisation and in relation to super-enhancers. Using these newly available data types has great potential to increase understanding of this process.

In this manuscript, the authors identify genes with recurrent HIV-1 insertions (RIGs) and demonstrate that such genes tend to have higher levels of active chromatin marks, higher expression, and be closer to super-enhancers even if they are not highly expressed. Using Hi-C data, the authors show that insertion sites have higher interactions between themselves than non-insertion sites, and identify an enrichment of insertion sites in the "A1" chromatin sub-compartment that they define based on Hi-C interactions. While the A1 compartment contains active genes, they find that increased insertions are not explained by higher gene expression in this compartment. They also look at the nuclear positioning of RIGs in resting and activated T cells, and suggest that super-enhancers play a role in movement of these genes towards the nuclear periphery upon activation.

Unfortunately, in this version of the manuscript, the main conclusions of the work are unclear: the authors find that insertion preference can be partially attributed to each of gene expression level, proximity to super-enhancers, and genomic sub-compartment, but they do not clearly establish the relative contributions of each of these features. In addition, the assertion that genes with super-enhancers move towards the nuclear periphery upon T cell activation is not sufficiently supported and seems disconnected from the other findings.

Major points

- The authors find that insertion preference cannot be attributed to any one of gene expression level, proximity to super-enhancers, or genomic sub-compartment, but the picture presented is confusing and doesn't clearly establish what the authors believe to be the main determinant of insertion preference, especially given the strong positive correlations between these features. Statistical modelling of the contributions of these features to insertion rate may help to clarify their relative contributions and the interactions between them (although the correlations between them might cause problems here).
- The Hi-C analysis approach is not sufficiently described. In particular, did the authors perform matrix normalisation using e.g. iterative correction or Knight-Ruiz matrix balancing?
- It is unclear why the authors define fifteen sub-compartments at the intra-chromosomal level and then reduce this to five sub-compartments when incorporating the inter-chromosomal data. Additional figures to illustrate the differences between sub-compartments and justify the number of different sub-compartments used would strengthen this analysis.
- The authors claim that RIGs and genes with super-enhancers become more peripheral upon T cell activation, but this statement is not sufficiently supported by the figures shown. First, RIGs without super-enhancers (MKL2 and TNRC6C) also seem to move to the periphery despite not having super-enhancers. Furthermore, the authors only show one example of a gene that remains in the centre of the nucleus upon T cell activation, which is odd considering this is presumably a large category (most active genes are not at the nuclear periphery). There is also only one example of a non-RIG gene with super-enhancers shown. Without seeing additional examples of non-RIG genes that do not move towards the nuclear periphery upon activation, I wonder if perhaps most active genes move towards the periphery, not just RIGs / genes with super-enhancers.
- The relevance of the T cell activation system to HIV-1 integration is not clear – can HIV-1 integrate into the genomes of in both resting and activated T cells, or is there a difference? More

background is required for publication in a journal with a broad audience.

- In addition, for the FISH data presented in Figure 6 of the manuscript, although there is a statistically significant change in the radial position between activated and resting T cells, the resulting position doesn't really correspond to the nuclear periphery, except maybe for NFAT3C, which already displays a peripheral localisation in resting T cells (in 10% of the nuclei examined).
- Throughout the analysis, the authors use a set of RIGs defined using a combination of patient data and in vitro data. However, in the discussion they state that the insertion patterns found may be due to certain insertion locations providing a selective advantage in patients. Therefore, would it not make sense to analyse separately insertion sites found in patients vs those found in in vitro experiments? This may resolve the question they pose in the final paragraph of the discussion, that is whether preferential insertion near super-enhancers is due to open chromatin at these regions or if it provides a selective advantage.

Minor points

- The ROC analysis is not well explained. Diagrams, perhaps in a supplementary figure, would be helpful here.
- The number of identified non-RIG genes (51939) is extremely high – have the authors excluded pseudogenes from this category? If not this may bias the results shown in Fig 1A, as pseudogenes should have low levels of active chromatin marks.
- The authors state that either ~14.5k, ~39k, or ~28k insertion sites were identified in primary CD4+ T cells, in different sections of the manuscript. Please explain this discrepancy.
- The authors use non-standard terminology in several places, which may hinder understanding of the manuscript by readers (“super-enhanced” to mean “putatively regulated by a super-enhancer”; “delineated with” to mean “proximal to”; “loading” of super-enhancers to mean “binding of BRD4 to super-enhancers”).
- “In contrast, the coverage of euchromatin marks and the transcriptional activity are only slightly higher in A1 than in A2 (Figures 5D and 5F).” This is misleading, as figure 5F y axis is log scale.
- The analysis of co-localisation of pairs of RIGs in Figure 6D-E doesn't seem to be connected to any of the major claims of the manuscript.
- “stretched enhancers” \diamond “stretch enhancers”
- “MLV” acronym is not defined

Reviewer #2 (Remarks to the Author):

In this study, Lusic and colleagues expand on their previous observation that HIV-1 preferentially integrates into genes in the nuclear periphery, in regions with active chromatin markers, near nuclear pore complexes. This article showcases a substantial amount of data: Hi-C in Jurkat cells, ChIP-Seq in primary CD4T cells, HIV integration mapping in CD4 T cells, multiplexed FISH assays in addition to an extensive mega-analysis of published data on HIV integration sites. The study shows a new connection of HIV integration with genes close to superenhancers and proposes the model that it is not the higher transcriptional activity as so far assumed but the nuclear location of these genes and possibly their movement from the nuclear core to the nuclear periphery during T cell activation that functions as decisive factor for integration. The manuscript is very data-dense including high level bioinformatics analysis. Major concerns include the changing experimental models (mega-analysis of patient data with clonally expanded cells, in vitro infected CD4+ T cells, Jurkat cell lines) - all lumped together with mostly cited evidence that they are comparable; but one wonders why not all experiments are done more tractably in Jurkat cells if they show the same phenotype as primary T cells. Mechanistic studies are confined to experiments with BET inhibitor JQ1 to alter superenhancer activity which surprisingly and somehow disappointingly did not show effects on integration preference or efficiency but showed effects on movement of superenhancer-driven genes, findings mechanistically hard to reconcile if the model is that outward movement during T cell activation is the decisive factor for recurrent integration. Overall an interesting

manuscript, but in the critical experiments not yet fully cohesive.

Key Claims

1. Recurrently targeted genes are delineated with super-enhancer genomic elements.
2. These genes are found in specific spatial compartments of the T cell nucleus
3. These gene clusters acquire the location during the activation of cells
4. The location/clustering of the genes along with the transcriptional status of the genes determines whether or not the virus integrates into these genomic elements.
5. These genomic regions get re-positioned upon T cell activation

Specific Comments

1. Figure 1C shows that the more databases identify a gene as integration site, the closer it is to a super enhancer (SE), however its likely that the numbers of genes identified in 1–7 data sets is reduced in the higher categories. How many genes fall into each category at each step? If the number is reduced, does that introduce a random bias towards landing in super enhancers?
2. Figure 1D, chr 6: integration site does not seem to match with super enhancer signals despite otherwise concluded in the manuscript. Please clarify!
3. Figure 2A-2C have confusing y-axis
 - a. What does regularized log transformed read counts mean?
 - b. In figure 2C it looks like genes near super enhancers are generally more highly expressed anyway; if integration is correlated with expression it would make sense that integration sites were near SEs (the more subtle difference in genes with 2 or more makes sense because the expression is already on average higher). Shouldn't there be a more broad gene expression profile if claims that expression did not matter were true? The subtle phenotype is acknowledge in the text. (Page 2)
4. JQ1 treatment should disrupt BRD4 binding to super enhancers and change the chromatin environment. Wouldn't this lead to changes in integration as a consequence? Figure 3 clearly shows that JQ1 does nothing (in line with other published results). Please comment and also reconcile with results in Figure 6. It would be relatively easy to determine what JQ1 in fact does to SE function and localization by including JQ1-treated cells in various assays especially the Hi-C.
5. Figure 5 again indicates that gene expression does matter for HIV-1 integration. In 5E, there is an enrichment of HIV-1 insertions in highly expressed genes - not a new observation, but well reproduced in these Hi-C data.
6. Figure 6: The finding that JQ1 does not affect T cell activation is surprising given published data on I-Bet suppressing T cell activation. Please comment. In addition, the FISH experiments are inconsistent in the phenotype in that some RIGs do only move upon activation, some don't, in some JQ1 has an effect preventing outward movement and in some it does not. Overall this observation and its relevance for HIV integration sounds more anecdotal than a consistent observation.
7. It is confusing that at the beginning the paper focuses on genes in the vicinity of SEs as integration sites, later on SE as integration sites themselves. Please clarify and add a model.

Response to the Reviewers by Lucic et al.

Reviewer #1:

Spatially clustered loci with multiple enhancers are frequent targets of HIV-1, Lucic et. al

This manuscript describes an analysis of HIV-1 integration patterns in the context of three-dimensional chromatin organisation and in relation to super-enhancers. Using these newly available data types has great potential to increase understanding of this process.

In this manuscript, the authors identify genes with recurrent HIV-1 insertions (RIGs) and demonstrate that such genes tend to have higher levels of active chromatin marks, higher expression, and be closer to super-enhancers even if they are not highly expressed. Using Hi-C data, the authors show that insertion sites have higher interactions between themselves than non-insertion sites, and identify an enrichment of insertion sites in the "A1" chromatin sub-compartment that they define based on Hi-C interactions. While the A1 compartment contains active genes, they find that increased insertions are not explained by higher gene expression in this compartment. They also look at the nuclear positioning of RIGs in resting and activated T cells, and suggest that super-enhancers play a role in movement of these genes towards the nuclear periphery upon activation.

Unfortunately, in this version of the manuscript, the main conclusions of the work are unclear: the authors find that insertion preference can be partially attributed to each of gene expression level, proximity to super-enhancers, and genomic sub-compartment, but they do not clearly establish the relative contributions of each of these features. In addition, the assertion that genes with super-enhancers move towards the nuclear periphery upon T cell activation is not sufficiently supported and seems disconnected from the other findings.

We very much appreciated the comments and suggestions made by this Reviewer, and we have tried to further improve our manuscript in the current revision by addressing all the Reviewer's points.

Major points

(1)• The authors find that insertion preference cannot be attributed to any one of gene expression level, proximity to super-enhancers, or genomic sub-compartment, but the picture presented is confusing and doesn't clearly establish what the authors believe to be the main determinant of insertion preference, especially given the strong positive correlations between these features. Statistical modelling of the contributions of these features to insertion rate may help to clarify their relative contributions and the interactions between them (although the correlations between them might cause problems here).

We are thankful to the Reviewer for their valuable comment. We have addressed this issue with logistic regression and have included the results in the new version of the manuscript. This in turn allowed us to clarify one of the main messages in a compact way, namely that genome organization marginally contributes to defining HIV target genes, but it does contribute to defining HIV hotspots.

As explained in detail in the manuscript, we have used the three variables above plus gene size to predict whether a gene is a “typical” HIV target (defined as top 33% highest HIV insertion rate), and to predict whether it contains a hotspot (defined as a 10 kb bin with 5 or more HIV insertions, *i.e.* top 2.5% genes with highest bin-wise insertion rate). To address the issue of feature redundancy and correlation, we have calculated the drop in accuracy incurred by removing the variable from the model. This measures the information that cannot be compensated by remaining variables. Gene expression is almost the sole contributor to HIV typical targets, whereas for HIV hotspots, with sub-compartments as the major contributor.

Those results are presented in **Figure 5H**. The variables do not necessarily provide a direct causal relationship with the HIV target, nevertheless the models establish that HIV hotspots are intrinsically distinct from typical targets because genome organization has a significant contribution to the former, but not to the latter.

(2)• The Hi-C analysis approach is not sufficiently described. In particular, did the authors perform matrix normalisation using e.g. iterative correction or Knight-Ruiz matrix balancing? We have clarified the description of the data processing in the Methods section. In the previous version of the manuscript, neither iterative correction (ICE) nor Knight-Ruiz balancing (KR) were used, because our pilot tests suggested that they had little impact on the results.

We have nevertheless decided to perform ICE normalization. We have also changed the weights of the eigenvectors to the square root of the associated eigenvalue (the previous weights were equal to the eigenvalues; by taking the square root, the weights can be interpreted as the projected variance as in Principal Component Analysis). It turned out that the weights have an impact on the segmentation into sub-compartments. Our main conclusions remain unaffected, but we would like to highlight the effects of this update for the sake of transparency. The Pie chart showing the coverage of the sub-compartments in the main figure 5B has now been changed.

Below is the difference of genomic coverage. The AB sub-compartment was substantially reduced in the update, to the benefit of A2 and B2. The B1 sub-compartment now occupies a very small portion of the genome.

Figure R1: Sub-compartments coverage using previous and current normalization.

Below is a bar plot showing the ChIP-Seq features present in the sub-compartments. The major difference is that B1 can now be interpreted as a H3K9me3 compartment. Also

observe that the H3K27me3 compartment has switched from B1 to AB and that Lamin is now high in both B1 and B2. The main figure 5D has been update accordingly.

Figure R2: ChIP-Seq features of the sub-compartments in the previous and current coverage.

Below is a box plot showing the expression of Jurkat genes. The major change is that the expressions of AB and B1 were swapped, consistently with the swap of the H3K27me3 mark. The main figure 5F has now been updated.

Figure R3: Expression of active genes in sub-compartments.

The changes are notable, but they have no consequences on the conclusions of the manuscript because they are based on the comparison between A1 and A2, which are

practically unaffected by the update (except the coverage of A2). This also means that the definitions of A1 and A2 are robust and invariate to the details of the implementation. AB, and B1 seem to be more affected, due to many possible ways to define them. We revised their definition because it corresponds to more widely accepted data processing steps. Even though we do not make direct use of AB, B1 and B2, we highlight in the Methods section that the definition of AB and B1 is somewhat arbitrary.

For completeness, we show below that balancing by ICE has little effect on the coverage of the sub-compartments. We observed similar results for ChIP-Seq features, gene expression and A/B scores. This means that the differences presented above are mostly due to the new weights used when projecting the matrices before clustering.

Figure R4: Comparison between no balancing and ICE normalization.

(3) It is unclear why the authors define fifteen sub-compartments at the intra-chromosomal level and then reduce this to five sub-compartments when incorporating the inter-chromosomal data. Additional figures to illustrate the differences between sub-compartments and justify the number of different sub-compartments used would strengthen this analysis.

We have improved the Hi-C analysis section with new supplementary figures and a description of the rationale behind the choice of the number of sub-compartments.

We chose the number of clusters based on previous clustering performed at much lower resolution where the authors identified 6 clusters (Rao et al. Cell 2014). Our initial choice ($k_1=15$) was to account for the possibility that we would discover new conformations visible only at high resolution (5 kb in our case). In most cases, however, the ChIP-seq / DamID data in the 15 intra-chromosomal clusters showed no more than 5 clearly individualized patterns.

Figure R5: The clustering heatmap of Chr1.

For instance, in the clustering heatmap of chromosome 1 above, clusters 0 and 14 have similar ChIP-seq / DamID features. The same goes for clusters 5 and 3, and for clusters 7, 6, 2, 12 and 9. Clusters 8, 1, 10 and 4 are also similar, but they are discounted to exclude non-mappable regions where all the features seem to be absent. Observe that those clusters correspond to three “heterochromatic” types, rich in H3K9me3, H3K27me3 or Lamin, and to two “euchromatic” types, rich or extremely rich in all the active chromatin marks.

We performed the same analysis with increasing number of intra-chromosomal (k_1) and inter-chromosomal (k_2) compartments and the result consistently showed repeated patterns of the five chromatin types, which suggested that the additional sub-compartments were merely subsets of the previous ones. This shows that the choice of k_1 has little influence, as long as it is greater than 5, and that the value $k_2=5$ imposes itself from the data.

(4)• The authors claim that RIGs and genes with super-enhancers become more peripheral upon T cell activation, but this statement is not sufficiently supported by the figures shown. First, RIGs without super-enhancers (MKL2 and TNRC6C) also seem to move to the periphery despite not having super-enhancers. Furthermore, the authors only show one example of a gene that remains in the centre of the nucleus upon T cell activation, which is odd considering this is presumably a large category (most active genes are not at the nuclear periphery).

There is also only one example of a non-RIG gene with super-enhancers shown.

Without seeing additional examples of non-RIG genes that do not move towards the nuclear periphery upon activation, I wonder if perhaps most active genes move towards the periphery, not just RIGs / genes with super-enhancers.

In order to address the concerns of the Reviewer, we have performed additional FISH experiments, and divided the NON-RIG controls into two groups:

1. Non-RIGs with SE,
2. Non-RIGs without SE (expressed genes).

1. NON-RIGs with SE

In addition to the typical super-enhancer gene *MYC*, we now examined the position of other 3 loci/genes

A) Chr 11q12.1 containing genes proximal to super-enhancers *SLC43A1*, *UBE2L6* and *TIMM10* (covered by BAC RP11-624G17); these expressed genes are not targeted by HIV-1. We observed movement towards the external regions, from zone 3 to zone 1 in activated cells. This behaviour is similar to the *MYC* gene.

B) *LMNA*, gene with a super-enhancer, also shows statistically significant repositioning, shifting from zone 3 towards zone 2.

C) *MSN* (Moesin) gene on Chromosome X, is an expressed gene with a super-enhancer. covered by BAC CH17-413H2. In this case the statistically significant repositioning occurs mostly from zone 2 to zone 1 of the nucleus.

Figure R6

Figure R6: Three-dimensional immuno-DNA FISH for control genes in resting and activated (anti-CD3/anti-CD28 beads, IL-2 for 48 h) CD4⁺ T cells (green: BAC/gene probe, red: lamin B1, blue: DNA staining with Hoechst 33342, scale bar represents 2 μm). Cumulative frequency plots show combined data from both experiments (n = 100, black: resting cells, red: activated cells). The p-values of the Kolmogorov-Smirnov tests are indicated. Bar plots represent percentages of alleles in nuclear zones, where 1 is peripheral, and 3 central. Signals with signal-to-radius ratios smaller than 0.19 were assigned to zone 1, signals with ratios ranging from 0.19 to 0.43 to zone 2, and signals with ratios higher than 0.43 to zone 3 (as in ¹). Representative images are shown for A) *SLC43A1*, *TIMM10* and *UBE2L6*, B) *LMNA* C) *MSN*.

The newly acquired FISH data were tested for significance of difference between resting and activated CD4⁺ with the Kolmogorov-Smirnov (KS) test. (graphs in the lowest panels). In addition, data were also represented as distribution of alleles in 3 different zones, as previously described in ¹. In these graphs shown in the middle, signals were distributed in 3 equal surface areas upon normalization over the nuclear radius (defined as half of the middle of the laminin B1-stained ring), followed by binning into three classes of equal surface area. Such graphical representation allows to visualize non random distribution of tested alleles vs random distribution (allele found with equal frequency (33%) in each of the three zones of the nucleus: 1-peripheral, 2-middle and 3-central).

When taking into consideration the results previously obtained for the *MYC* gene, it can be concluded that genes proximal to super-enhancers show statistically significant repositioning towards the outer shells of the nucleus.

This is very different from what we obtained for the second group of tested genes, that we analysed following Reviewers comments genes that are expressed but do not have SE elements.

2. Non-RIGs with no SE

To fully address Reviewers comments related to the general spatial “behaviour” of expressed genes, we analysed a second group of control expressed genes that do not have SE elements (and are not targeted by HIV-1).

One such gene is *TAP1* gene, which was shown in the Supplementary Figure 6B in the original submission. Following Reviewers comments, we now analysed 4 additional genes in regions: *MCM4* gene on Chr 8 (Figure R7A), *CCNC* on Chr 6 (Fig R7B), *RECQL* (Fig R7C) on Chr 12 and *MARCH1* on Chr4 (Fig R7D). These genes are expressed in activated CD4⁺ T cells, do not have SE and are not found to be targeted by HIV-1. None of them showed statistically significant repositioning in the nuclear space during T cell activation. Interestingly, two genes *RECQL* and *MARCH1* were found in the outer shells of the nucleus of both resting and activated T cells (preferential distribution in zone 1), and are not repositioned in the nuclear space.

Figure R7

Figure R7: Three-dimensional immuno-DNA FISH for four control genes in resting and activated (anti-CD3/anti-CD28 beads, IL-2 for 48 h) CD4⁺ T cells (green: BAC/gene probe, red: lamin B1, blue: DNA staining with Hoechst 33342, scale bar represents 2 μ m). Cumulative frequency plots show combined data from both experiments (n = 100, black: resting cells, red: activated cells). The p-values of the Kolmogorov-Smirnov tests are indicated. Bar plots represent percentages of alleles in nuclear zones, where 1 is peripheral, and 3

central. Signals with signal-to-radius ratios smaller than 0.19 were assigned to zone 1, signals with ratios ranging from 0.19 to 0.43 to zone 2, and signals with ratios higher than 0.43 to zone 3 (as in ¹). Representative images are shown for A) *MCM4*, B) *CCNC*, C) *RECQL* and D) *MARCH1*.

In summary, our data show that a group of genes with super-enhancers move towards the outer shells of the nucleus during T cell activation (see also point 6 and data in Figure R8). This is not a general feature of all active genes, as examined genes without super-enhancers do not change their position during T cell activation.

As pointed out by the Reviewer, exception to this rule are genes *TNRC6C* and *MKL2* that do not have super-enhancers but still move towards the periphery.

TNRC6C gene is part of a clustered region of 2.6 MB, containing RIGs proximal to super-enhancers *GRB2* and *RNF157*. **Figure 6E** shows that they cluster together, while the FISH data in **Figure 6C** show repositioning towards the nuclear periphery for each of them. We presume that *TNRC6C* does not move independently, but migrates as part of a cluster, together with its neighboring genes *GRB2* and *RNF157*, both proximal to super-enhancers. *MKL2*, the other example of a RIG without a SE, does not have neighboring genes with SE that could foster its repositioning towards the outer shells of the nucleus.

While we cannot exclude that other examples could dispute our conclusions, data obtained so far suggest that a statistically significant fraction of genes proximal to super-enhancers change their spatial distribution in response to T cell activation. In the future study we plan to perform a robust labeling and imaging analysis to define the spatial positioning of the larger number of genes with or without super-enhancers.

(5)• The relevance of the T cell activation system to HIV-1 integration is not clear – can HIV-1 integrate into the genomes of in both resting and activated T cells, or is there a difference? More background is required for publication in a journal with a broad audience. We thank the reviewer for this comment, we have modified the main text to better explain this point.

(6)• In addition, for the FISH data presented in Figure 6 of the manuscript, although there is a statistically significant change in the radial position between activated and resting T cells, the resulting position doesn't really correspond to the nuclear periphery, except maybe for *NFAT3C*, which already displays a peripheral localisation in resting T cells (in 10% of the nuclei examined).

The Reviewer is correct, there are only few examples of genes that are indeed peripheral, *NFAT3C* being one of them. Other examples of peripheral RIGs include *BACH2*, *RPTOR* and *NPLOC4*, but these genes do not change their radial position during T cell activation. To address this concern, we looked specifically at HIV-1 RIGs that move: *FOXP1*, *STAT5B*, *NFAT3C*, *MKL2*, *KDM2A*, *PACS*, *GRB*, *RNF* and *TNRC6C* (shown in Figure 6A-C). We plotted the frequency distribution for all these alleles in resting cells (n=1700), and compared with their distribution in activated cells (n=1690 alleles) in 3 zones of the nucleus. These 3 zones of the equal surface areas, were obtained after normalization over the nuclear radius (defined as half of the middle of the lamin B1-stained ring), followed by binning into three classes of equal surface area. While graphs in B show distributions of alleles in 3 zones separately for resting and activated cells, the graph in panel A nicely demonstrates the shift of distribution towards zone 1 (less than 1 micron under the nuclear envelope) in activated T cells. This analysis has now been added to the main text (**Figure 6D**).

Figure R8: Allele fraction density plot for all resting and activated alleles that displayed peripheral repositioning. Y axis shows allele fraction density for all measured genes shown in Figure 6A (*FOXP1*, *STAT5B*, *NFATC3*, *MKL2*, *KDM2A*, *PACS*, *GRB*, *RNF* and *TNRC6C*). X axis represents ratios of distance from nuclear envelope (lamin B1 staining) and radius (signal to radius ratio) for all these alleles in resting cells (n=1700), and activated cells (n=1690 alleles). Binning into equal 3 concentric zones of the nucleus is performed as in ¹.

(7)• Throughout the analysis, the authors use a set of RIGs defined using a combination of patient data and in vitro data. However, in the discussion they state that the insertion patterns found may be due to certain insertion locations providing a selective advantage in patients. Therefore, would it not make sense to analyse separately insertion sites found in patients vs those found in in vitro experiments? This may resolve the question they pose in the final paragraph of the discussion, that is whether preferential insertion near super-enhancers is due to open chromatin at these regions or if it provides a selective advantage.

We agree with the Reviewer that it would be justified to look at patient data only. Unfortunately, defining the chromatin profiles/genomic organization/spatial genome distribution in those samples is not feasible. However, it was possible to analyse separately insertion sites found in in vitro experiments (Brady and Lucic data sets) vs those found in patients (other experiments). This comparison was done on the level of integration sites and on the level of genes. For that reason we compared patient data with integration sites in primary infection, defined the chromatin profiles and then analyzed chromatin mark distributions on the integration sites separately for each data set (study). In **Figure 1B** we show co-occurrence of integration sites and epigenetic modifications obtained by ChIP-Seq for H3K27ac, H3K4me1, BRD4, MED1, H3K36me3, H4K20me1, H3K4me3, H3K27me3 and H3K9me2, for all data sets separately. Columns marked as Brady and Lucic correspond to *in vitro* experiments while the rest of the columns show results from patients divided in different lists. We did not observe separation of clusters of patient-derived data vs. data sets obtained *in vitro*.

When integration preferences are analysed on the level of genes, again no separation of patient vs *in vitro* data sets can be observed. This conclusion is reached when looking at the percentage of overlaps between all pairs of data sets. The following heatmap (Figure R9) shows hierarchical clustering of data sets based on their pairwise similarities. Each cell

represents fraction of genes found in data set X (rows) that is also found in data set Y (columns). Note that this matrix is not diagonally symmetric due to different sizes of the data sets (% of A in B is different from %B in A). When applying hierarchical average linkage clustering, we observe that smallestRandom and largestRandom mock data sets cluster together as expected and that there is no separation of patient and *in vitro* data sets. We decided to show supplementary figure 3 instead of this figure due to simplicity. Supplementary figure 3 shows the fraction of genes from each data set that is shared with at least one other data set.

We conclude that there is no difference in integration site preferences in data sets obtained from patients and the ones obtained from *in vitro* experiments. This is expected due to relatively low number of clonally expanded sites in the experiments where this was tracked (From 1632 different integration events found in ²,1388 were found only once while 244 were clonally expanded). Thus, to analyse evolutionary advantage of site selection we would need to obtain more integration data sets from longitudinal experiments, which lack sufficient data sets and would in any case reach beyond the scope of this work..

The point we wanted to make in the discussion is that *in vitro* integrations do not show different preferences from the ones obtained from patients and therefore can be used to understand how clonally expanded sites could arise under the selective pressure of current therapies.

Figure R9: Heatmap of pairwise similarities between data sets. Each cell represents fraction of unique genes overlapping integrations from data set in the column with unique genes overlapping integrations in data set in the row. Data sets are clustered using hierarchical clustering with average linkage as a method to calculate distances between clusters. Data sets marked as Brady and Lucic correspond to *in vitro* experiments while the rest of the data sets originate from patients,. smallestRandom and largestRandom represent control (mock) data sets.

Minor points

- The ROC analysis is not well explained. Diagrams, perhaps in a supplementary figure, would be helpful here.

In order to assess if there is an enrichment of various chromatin features on sites of HIV-1 integration, we adapted the ROC curve areas method from (Berry et al. 2006). In short, the strategy was to use “nested case controls” - a collection of integration sites sampled from the genome which would act as control sites and can be compared to true integration sites. Those control sites were generated to account for bias of integration towards genes - they were sampled to match true sites in distance to nearest gene. For each true integration site, we generated 10 matched control sites and compared various chromatin features of the matched sites with the chromatin features of the true site. For each true integration site and chromatin feature, we counted a fraction of control sites having a lower feature value (e.g. true site has higher H3K27ac value than n percent of control matched sites). We averaged the results over all integration sites within data sets. This procedure is explained in details in supplementary Text S1 of the cited paper: <https://doi.org/10.1371/journal.pcbi.0020157.sd001> and in the Supplementary Text S3 from the ³ Inference for ROC curve areas for Genomic Features using Matched Random Controls. We updated the methods to explain this analysis better.

- The number of identified non-RIG genes (51939) is extremely high – have the authors excluded pseudogenes from this category? If not this may bias the results shown in Fig 1A, as pseudogenes should have low levels of active chromatin marks.

We apologize if this was not clear from the original submission. We have updated the figure legends, main text and methods to state this clearly.

In those cases where using all genes could have introduced bias to the results (for ChIP-Seq profiles on genes and expression data), only protein coding genes were used for the analysis. This was the case in the following figures: **Figure 1A, Figure 2A, 2B, 2C, 2D, Figure 3D, Figure 5F, Supplementary figure 2 (B, C) and Supplementary figure 5B.** List of all genes from Ensembl, GRCh37 assembly from February 2014 was used for the **Figures 1C, Supplementary Figure 1 (A, B, C, D) and Supplementary Figure 3.**

- The authors state that either ~14.5k, ~39k, or ~28k insertion sites were identified in primary CD4+ T cells, in different sections of the manuscript. Please explain this discrepancy.

We apologize for not clarifying this in the first instance, the information is now added to the methods, section Bioinformatic analysis, paragraph on integration site analysis.

- The authors use non-standard terminology in several places, which may hinder understanding of the manuscript by readers ("super-enhanced" to mean "putatively regulated by a super-enhancer"; "delineated with" to mean "proximal to"; "loading" of super-enhancers to mean "binding of BRD4 to super-enhancers").

We are thankful to the Reviewer for their comment, and we have now modified the terminology throughout the text.

- "In contrast, the coverage of euchromatin marks and the transcriptional activity are only slightly higher in A1 than in A2 (Figures 5D and 5F)." This is misleading, as figure 5F y axis is log scale.

The Reviewer is correct, it is indeed a log scale, and this information has now been added to the figure legend. To clarify the difference in enrichment we also state in the text the A1/A2

ratios for median gene expression (1.12), chromatin coverage (1.04 for H3K27Ac and 1.09 for H3K36me3) and HIV-1 density (2.71).

- The analysis of co-localisation of pairs of RIGs in Figure 6D-E doesn't seem to be connected to any of the major claims of the manuscript.

The purpose of these experiments was to assess clustering of HIV-1 RIGs both in resting and activated cells. As shown in FISH analysis for each of the genes separately (**Figures 6B** and **6C**), these genes change their radial position in activated cells. Figure 6D-E and in revised version of the manuscript Figure 6E and F show that they also cluster together in both resting and activated cells, thus it can be concluded that they move together i.e. in clusters towards periphery.

These examples are further supporting the notion that HIV-1 targets clusters, rather than single genes, which are repositioned towards the outer shells of the nucleus during T cell activation.

The main text was modified, and we hope that the new modified text serves to explain better the purpose of this experiment.

- "stretched enhancers" "stretch enhancers" This was changed.
- "MLV" acronym is not defined This was now defined.

Reviewer #2:

In this study, Lusic and colleagues expand on their previous observation that HIV-1 preferentially integrates into genes in the nuclear periphery, in regions with active chromatin markers, near nuclear pore complexes. This article showcases a substantial amount of data: Hi-C in Jurkat cells, Chip-Seq in primary CD4T cells, HIV integration mapping in CD4 T cells, multiplexed FISH assays in addition to an extensive mega-analysis of published data on HIV integration sites. The study shows a new connection of HIV integration with genes close to superenhancers and proposes the model that it is not the higher transcriptional activity as so far assumed but the nuclear location of these genes and possibly their movement from the nuclear core to the nuclear periphery during T cell activation that functions as decisive factor for integration. The manuscript is very data-dense including high level bioinformatics analysis. Major concerns include the changing experimental models (mega-analysis of patient data with clonally expanded cells, in vitro infected CD4+ T cells, Jurkat cell lines) - all lumped together with mostly cited evidence that they are comparable; but one wonders why not all experiments are done more tractably in Jurkat cells if they show the same phenotype as primary T cells.

Mechanistic studies are confined to experiments with BET inhibitor JQ1 to alter superenhancer activity which surprisingly and somehow disappointingly did not show effects on integration preference or efficiency but showed effects on movement of superenhancer-driven genes, findings mechanistically hard to reconcile if the model is that outward movement during T cell activation is the decisive factor for recurrent integration. Overall an interesting manuscript, but in the critical experiments not yet fully cohesive.

Key Claims

1. Recurrently targeted genes are delineated with super-enhancer genomic elements.

2. These genes are found in specific spatial compartments of the T cell nucleus
3. These gene clusters acquire the location during the activation of cells
4. The location/clustering of the genes along with the transcriptional status of the genes determines whether or not the virus integrates into these genomic elements.
5. These genomic regions get re-positioned upon T cell activation

We are grateful to the Reviewer for appreciating our work and we are happy to provide additional data as well as a better explanation for the choice of the experimental models. We are aware of the confusion caused by changes in experimental models: ideally we would have preferred to perform all the analyses on one type of cells but this is impossible because Jurkat cells represent activated T cells (main targets of new HIV-1 infections), but not resting T cells of the memory phenotype (main reservoir).

We started by looking at patient data, where limited amounts of available material preclude chromatin analysis such as ChIP-Seq or Hi-C. We therefore opted for two closest models: *in vitro* infections of either primary CD4⁺ T cells or Jurkat cell line.

Jurkat cells are indeed a very useful tool for studying HIV-1 infection. As a transformed cell line, Jurkats proliferate constantly and can readily be infected with the virus (without any previous stimulation). They therefore resemble to the phenotype of activated, rather than to the resting CD4⁺ T cells. In line with this, the size of the Jurkat nucleus corresponds to the one of activated cells, while it is significantly bigger than the nucleus of resting CD4⁺ T cells (table in Figure R10A). Moreover, when we checked for the presence of an early T cell activation marker CD69, which is absent from resting primary cells and detectable on CD4⁺T cells only after activation (with CD3/CD28 antibody coated beads), we observed that Jurkat cells have some sort of an intermediate phenotype: the marker is present in 63% of the non-activated Jurkat cells, but its amounts can further be increased to 69% upon exposure to the activation stimuli (compare resting in black vs activated in green) (Figure R10B).

Figure R10: RIGs proximal to super-enhancers positioning in Jurkat and CD4⁺ T cells:

A) Cell diameter (μm) for Jurkat and T cells FISH measurements obtained above. B) FACS histogram of Jurkat cells before and after cell activation with anti-CD3/anti-CD28 beads, IL-2 for 24 h. Three-dimensional immuno-DNA FISH of three RIGs in Jurkat cells and resting and activated (anti-CD3/anti-CD28 beads, IL-2 for 48 h) CD4^+ T cells. Green: BAC/gene probe, red: lamin B1, blue: DNA staining with Hoechst 33342, scale bar represents 2 μm). Bar plots represent the distribution of the analyzed alleles into the three nuclear zones is depicted (as in ¹). C) *BACH2* D) *GRB2* and E) *PACS1* in Jurkat (left) and T cells (right). Error bars are shown for T cells (different donors), and not for Jurkats, as it is a cell line.

The same intermediate phenotype was observed in FISH, when HIV-1 target genes were compared between Jurkats cells and resting vs. activated T cells. We analysed *BACH2*, *GRB2* and *PACS1* genes as representative RIGs, and observed that Jurkat cells display an intermediate radial allele positioning phenotype (between resting and activated CD4^+ T phenotypes). While *BACH2* alleles localize peripherally in zone 1 in all cases (Figure 10RC), spatial distribution of *GRB2* and *PACS1* alleles in Jurkat cells is an intermediate between resting and activated T cell phenotypes (Figure 10RD and 10RE), resembling more the activated phenotype for *PACS1*.

For all these reasons, Jurkat cells can easily be infected with HIV-1 with the patterns of integration corresponding to the one of primary infections. These cells are a valuable tool for the studies of genome organization in Hi-C, especially considering that multiple chromatin marks assessed in ChIP-Seq are available for comparison, as we did here.

However, it was also very important to directly compare the spatial genome distribution of HIV-1 target genes in primary resting vs. activated T cell, which cannot be done on Jurkat because they are never resting (see above). Therefore our FISH analysis was performed on T cells, as well as the ChIP and RNA-Seq experiments.

The relevance of resting T cells for HIV-1 infection, as explained also in the newly added paragraph in introduction and in point nr.5 in response to Reviewer 1, is that these cells represent the main reservoir of latent HIV-1, but despite numerous studies performed on them, including our own previous work ^{4,5}, we still don't understand how these reservoirs are established. In comparison to activated CD4^+ T cells, in resting T cells multiple factors limit HIV-1 infection at both pre-integration and integration levels. According to current ideas ^{6,7} some of the activated T cells revert to the resting state upon infection, and the transition from one state to the other involve multiple changes in the chromatin structure and genome organization, as also documented in the present manuscript.

Of note, the first Hi-C map of primary human resting and activated CD4^+ T cells was published while we were revising this manuscript ⁸.

Specific Comments

1. Figure 1C shows that the more databases identify a gene as integration site, the closer it is to a super enhancer (SE), however its likely that the numbers of genes identified in 1–7 data sets is reduced in the higher categories. How many genes fall into each category at each step? If the number is reduced, does that introduce a random bias towards landing in super enhancers?

The following table shows number of genes in each category:

Number of lists a gene is found to have HIV integration in	0	1	2	3	4	5	6	7
Number of genes in this category	51939	3993	1102	431	191	80	18	8

To test if random bias towards landing into super-enhancers is introduced with reducing number of genes per category, we randomly selected 10 subsets of 50 genes from each of the categories (0 - 5) and made a boxplot of their distances to the nearest super-enhancer. “True” distance distribution is displayed for comparison for all genes in each respective category.

Figure R12: Boxplots showing distance to nearest super-enhancer for 10 random equally sized (n=50) subsets of genes divided by number of datasets they appear in as HIV-1 targets. Last boxplot of the group represents distances to nearest super-enhancers for all genes from a group. The upper whisker extends from the hinge to the largest value no further than 1.5 * IQR from the hinge (where IQR is the inter-quartile range, or distance between the first and third quartiles). The lower whisker extends from the hinge to the smallest value at most 1.5 * IQR of the hinge.

From this plot it is visible that means of all distances from 50 genes in a subset to the nearest super-enhancer vary more when genes are selected from a category with more genes than from the smaller one. In conclusion, from the first plot it is visible that size of category did not introduce a bias toward landing into super-enhancers. When selecting larger subsets (500 or 5000 genes), the difference among means stabilizes, which is why we decided to show boxplots for all possible genes from all categories, as shown in plot below for 10 subsets of different sizes drawn from category containing most genes:

Figure R13: Boxplots showing distance to nearest super-enhancer for 3 x 10 random equally sized (n=50, n=500, n=5000) subsets of genes that are not HIV targets. Last boxplot of the group represents distances to nearest super-enhancers for all genes that are not HIV-1 targets in any data sets. The upper whisker extends from the hinge to the largest value no further than 1.5 * IQR from the hinge (where IQR is the inter-quartile range, or distance between the first and third quartiles). The lower whisker extends from the hinge to the smallest value at most 1.5 * IQR of the hinge.

2. Figure 1D, chr 6: integration site does not seem to match with super-enhancer signals despite otherwise concluded in the manuscript. Please clarify!

The Reviewer is correct that the bulk of integration sites do not match the super-enhancer signal in the *BACH2* gene, but the **Figure 1D** demonstrates they are immediately adjacent to it. While for the other two examples, *FOXP1* and *STAT5B*, insertions are indeed inside the super-enhancer signals, in case of the *BACH2* example gene, integration sites cluster immediately next to it. The similar situation is observed for many other HIV-1 genes and is summarized in the table in Figure 1C.

As our main text says that HIV-1 integrates near the super-enhancers, we reasoned that this explanation could account for both scenarios. We hope that the Reviewer will agree on this.

3. Figure 2A-2C have confusing y-axis

a. What does regularized log transformed read counts mean?

The Reviewer has a point that this new transformation we used for RNA-Seq should have been explained more in detail, and we have included this in the methods part. We used a regularized logarithm transformation (rlog) to make our data more homoskedastic. The rlog transformation produces log₂ scale transformed data which has been normalized with respect to library size. We used rlog instead of log₂ transformation because it is more robust in the case when the size factors vary widely; this transformation is reducing the variance to avoid that the result becomes dominated by highly expressed, highly variable genes (original count scale data), or low expressed genes (if logarithm-transformed data are used). We

used a widely utilised rlog approach of DESeq2, which enables transformation similar to a log2 for genes with high counts, and resolves the problem for genes with low counts by compressing together the values for different samples. Otherwise, a standard logarithm transformation would spread apart the data, ie random noise could overtake the real biological signal⁹. We updated the methods with this explanation.

b. In Figure 2C it looks like genes near super-enhancers are generally more highly expressed anyway; if integration is correlated with expression it would make sense that integration sites were near SEs (the more subtle difference in genes with 2 or more makes sense because the expression is already on average higher). Shouldn't there be a more broad gene expression profile if claims that expression did not matter were true? The subtle phenotype is acknowledge in the text. (Page 2)

We do not claim that expression does not correlate with integration patterns. On the contrary, it has previously been shown that HIV targets active genes and this is confirmed in our paper as well.

This is why, to distinguish between effect of expression and proximity of super-enhancers, we looked separately into 4 groups of genes: 1) genes which are not expressed 2) genes which are in the 10% of low expressed genes, 3) intermediately expressed genes, 4) genes which are in the top 10% of expressed genes. These data shown in figure 2D demonstrate that in all categories, and independent on their expression, RIGs have a higher percentage of genes with super-enhancers in their proximity. We also constructed a model to explain the influence of expression, proximity of super-enhancers and chromatin structure in Jurkat cells on integration, that might serve to further clarify this matter.

4. JQ1 treatment should disrupt BRD4 binding to super enhancers and change the chromatin environment. Wouldn't this lead to changes in integration as a consequence? Figure 3 clearly shows that JQ1 does nothing (in line with other published results). Please comment and also reconcile with results in Figure 6. It would be relatively easy to determine what JQ1 in fact does to SE function and localization by including JQ1-treated cells in various assays especially the Hi-C.

The Reviewer is correct that JQ1 treatment should disrupt the binding of BRD4 to enhancers and should change the chromatin environment. It is nevertheless clear, as pointed out by the Reviewer, that the direct chromatin effects of JQ1 have no influence on HIV-1 insertion. In activated CD4⁺ T, JQ1 treatment had no effect on HIV-1 insertion, nor did it influence the spatial position of genes (**Fig 3C**) in activated CD4⁺ T cells. Performing a Hi-C experiment in JQ-1 treated and untreated cells is straightforward, but this does not directly address the question related to HIV-1 integration. The reason for this lies in the fact that JQ1 had an effect on spatial distribution of genes proximal to super-enhancers when resting T cells were pretreated with it (JQ1) before activation (**Supplementary Figures 6G** and **6H**). As resting T cells cannot efficiently be infected with HIV-1, the Hi-C experiment on such cells, apart from being very costly, would not in fact contain a corresponding set of HIV-1 integration sites to be compared to.

The FISH experiments we performed logically demonstrate that JQ1 affects the insertion site of HIV-1 if and only if it prevents the relocation of super-enhancers, in line with our main conclusion that super-enhancers are preferentially targeted because of their location rather than their chromatin context or activity. Additional Hi-C experiments would certainly shed

light into the effects of JQ1 on enhancer activity in different conditions, but they are unlikely to affect our main conclusions.

5. Figure 5 again indicates that gene expression does matter for HIV-1 integration. In 5E, there is an enrichment of HIV-1 insertions in highly expressed genes - not a new observation, but well reproduced in these Hi-C data.

As mentioned previously, we do not claim that expression does not correlate with integration patterns. This is why in supplementary figure 5B, we analysed density of HIV integration into different compartments, with respect to expression. When controlling for expression, we again see that there is a 2.5-fold enrichment of HIV integration into A1 compartment, indicating that HIV is intrinsically more frequent in the A1 compartment. To better understand relative contributions of expression, 3D compartments and proximity to super-enhancers to density of HIV integration to genes, we constructed a model and added explanation to the manuscript. In short, our conclusion is that gene expression is almost the sole contributor to determining HIV typical targets, whereas the picture is mixed for HIV hotspots, sub-compartments being the major contributor.

6. Figure 6: The finding that JQ1 does not affect T cell activation is surprising given published data on I-Bet suppressing T cell activation. Please comment.

In addition, the FISH experiments are inconsistent in the phenotype in that some RIGs do only move upon activation, some don't, in some JQ1 has an effect preventing outward movement and in some it does not. Overall this observation and its relevance for HIV integration sounds more anecdotal than a consistent observation.

To address the comment on the effect of Bet inhibitor i.e. JQ1 we would like to highlight that we performed several controls during our experimental setup. In particular, as part of the experiment in which we pretreated CD4⁺ T cells with JQ1 before activating then with anti-CD3/anti-CD28/ IL-2 (**Supplementary Figure 6G** and **6H**) we analysed the early (CD69) and late (CD25) T cell activation markers by flow cytometry. The early activation marker CD69 was detected in 63% of cells after 20 h of induction (Figure R14A left panel), with comparable numbers obtained also in JQ1-treated cells, where 65% of cells were found CD69-positive (**Figure R14A**). We measured 49% positive cell for CD25 and obtained almost identical results for JQ1 treated cells, as 50% of cells were positive for CD25 (**Figure R14B**). In summary, the similar kinetics of CD69 and CD25 upregulation upon activation was observed for both activated and JQ1-pretreated cells, indicating that JQ1 pretreatment does not generally impact T cell activation.

We also checked the expression levels of T cell activation markers in T cells treated with JQ1. In particular, we checked the following activation markers in our RNA-Seq data (resting cells vs resting + JQ1):

- 1) **CD69**, log₂FC=-0.33, p adjusted =0.39 (not significant)
- 2) **CD25 (IL2RA gene)** log₂FC=-0.89, p adjusted=0.06 (not significant)
- 3) General activation marker **CD38** - log₂FC=-0.89, p adjusted =0.02
- 4) Human leukocyte antigen DR (**HLA-DR HLA-DRA, HLA-DRB1 and HLA-DRB5**) not significant

In line with others, we did not observe a global effect of JQ1 on T cell activation, which is one of the reasons why JQ1 was proposed to be a promising latency reversing reagent (LRA)¹⁰.

Figure R14: Activation markers expression in CD4⁺ cells after JQ1 treatment and subsequent activation with anti-CD3/anti-CD28 beads and IL-2: A) CD69 and B) CD25 expression in DMSO-treated control cells (left) and JQ1 treated cells (right).

Regarding the second part of the Reviewer’s comment about FISH experiments, we would like to summarize here our finding:

We tested 9 RIGs in FISH (**Figure 6 A, B and C**) that all showed a statistically significant outward movement, and 3 RIGs (**Supplementary figure 6C**) that did not show any repositioning, as they were peripheral in both resting and activated cells.

Since almost all of the RIGs that displayed peripheral movement phenotype upon T cell activation have super-enhancers in their vicinity, and recent studies place SE elements at the nuclear periphery, possibly anchored to the NPC, we reasoned that their disassembly by JQ1 might compromise the peripheral movement phenotype occurring during T cell activation (**Figure 6**). Our data indeed suggest that this is the case as the effect of JQ1 was observed when resting cells were pretreated with the drug before activation, as explained in the paragraph above. In those conditions, two RIGs proximal to super-enhancers, which in activated cells reposition towards the periphery, were retained in their “resting” position (**Supplementary Figure 6G and 6H**).

Although the conventional FISH analysis can only be applied to a limited number of genes, we believe that the revised data, with addition control experiments, strongly support the notion that genes proximal to super-enhancers reposition in the 3D nuclear space in response to the activation signals. Considering that this is pertinent to all the tested genes proximal to super-enhancers, and not limited to HIV-1 RIGs, we plan to perform a more robust high throughput FISH analysis using large number of probes to fully address these interesting questions about the spatial distribution of genes proximal to super-enhancers.

7. It is confusing that at the beginning the paper focuses on genes in the vicinity of SEs as integration sites, later on SE as integration sites themselves. Please clarify and add a model. We apologize for the confusion. As we defined a gene to be “proximal” to a super-enhancer in case it contains one inside the body of the gene or at the distance of 5 kb upstream of its

transcription start site (Methods section, Super-enhancer calling) we used throughout the text different terms. We have now used only one term - integration into genes proximal to super-enhancer element(s).

As for the model, we were unsure which kind of model would the Reviewer like to see here: A statistical modelling, using different parameters that influence HIV-1 insertion sites (gene expression, vicinity to super-enhancers and gene clustering) was suggested by Reviewer 1, and this type of model has been added to the main Figure 5 and explained in point 1. We also prepared a graphical model, and added it here in response to the Reviewer. Should the Reviewer find this model useful, we can add it into Supplementary Figure 7 and add a paragraph into discussion to summarize it.

Figure R15: Model of HIV-1 integration into the 3D clusters of genes proximal to super-enhancers in activated CD4+ T cells. HIV-1 integration occurs rarely in resting CD4+ T cells that have large heterochromatin regions and few active genes. HIV-1 efficiently integrates in activated CD4+ T cells, where 3D clusters of active genes (Recurrent Integration Genes), often proximal to super-enhancers repositioned towards the outer shells of the nucleus, represent insertion hot-spots.

1. Marini, B. *et al.* Nuclear architecture dictates HIV-1 integration site selection. *Nature* **521**, 227–231 (2015).
2. Maldarelli, F. *et al.* HIV latency. Specific HIV integration sites are linked to clonal expansion and persistence of infected cells. *Science* **345**, 179–183 (2014).
3. Brady, T. *et al.* Integration target site selection by a resurrected human endogenous retrovirus. *Genes Dev.* **23**, 633–642 (2009).
4. Manganaro, L. *et al.* Concerted action of cellular JNK and Pin1 restricts HIV-1 genome integration to activated CD4+ T lymphocytes. *Nat. Med.* **16**, 329–333 (2010).
5. Lusic, M. *et al.* Proximity to PML nuclear bodies regulates HIV-1 latency in CD4+ T cells. *Cell Host Microbe* **13**, 665–677 (2013).
6. Sengupta, S. & Siliciano, R. F. Targeting the Latent Reservoir for HIV-1. *Immunity* **48**, 872–895 (2018).
7. Churchill, M. J., Deeks, S. G., Margolis, D. M., Siliciano, R. F. & Swanstrom, R. HIV reservoirs: what, where and how to target them. *Nat. Rev. Microbiol.* **14**, 55–60 (2016).
8. Gate, R. E. *et al.* Genetic determinants of co-accessible chromatin regions in activated T cells across humans. *Nat. Genet.* **50**, 1140–1150 (2018).
9. Love, M. I., Huber, W. & Anders, S. Moderated estimation of fold change and dispersion for RNA-seq data with DESeq2. *Genome Biol.* **15**, 550 (2014).
10. Banerjee, C. *et al.* BET bromodomain inhibition as a novel strategy for reactivation of HIV-1. *J. Leukoc. Biol.* **92**, 1147–1154 (2012).

Reviewers' comments:

Reviewer #1 (Remarks to the Author):

Overall, the authors have responded in a satisfactory way to many of the suggestions in the previous review. The revised manuscript is clearer and the background and context are described in more detail, enabling readers to better appreciate the significance and relevance of the results. However I have a few remaining concerns that should be addressed before I'd support publication of this manuscript.

Major concerns:

1) Thank you to the authors for including statistical modelling of the contributions of different factors to HIV integration. This definitely has potential to clarify the results of the manuscript and they key features affecting HIV integration. However, I have some concerns about the choice of features to predict and their relation to the other analyses.

a. In the statistical modelling, the authors choose to predict "typical targets" and "hotspots". These definitions are not used elsewhere in the manuscript, which makes it difficult to interpret the modelling results in relation to the other analyses. How do these two classes relate to each other and to the previously defined RIGs?

b. In particular, it's not clear why the authors chose to consider insertion rate normalised to gene length in order to define "typical targets" here, rather than just the total number of insertions per gene as used to define RIGs.

c. The "hotspots" are also not sufficiently described. They appear to be defined as 10kb bins, but then they should all have the same size, so why is "size" a factor in the model? Are the hotspots primarily found in non-coding regions, or do the authors only consider those that overlap with genes?

d. Most importantly, are the results of the modelling analysis robust to choosing different thresholds for the definitions of typical targets and hotspots? The most convincing analysis here would be to show that RIGs can be predicted, rather than adopting new definitions solely for this analysis.

2) Thanks also for the clarification of the Hi-C analysis pipeline that was used to produce the compartment assignments. Was the normalised Hi-C data also used for the interchromosomal interaction strength analyses presented in Figure 4? It is vital that normalised data is used here, as similar results would be produced if biases in the Hi-C procedure led to increased raw interaction

strengths at SEs and/or active genes (due to accessibility, GC content, mappability, etc...). Figure 4C has low image quality, but appears to show uneven “stripes” in the Hi-C matrix which are characteristic of unnormalised Hi-C data, leading me to be concerned about the processing of the data used for the analyses shown in this figure.

Minor concerns:

3) The calculation of the “ROC area under curve” values is still not sufficiently explained. Typically, ROC analysis involves plotting true positive rate vs false positive rate for varying parameter sets or thresholds. The analysis as described in the methods does not appear to relate to this typical ROC analysis, and I was unable to find any further clarification of this in the references provided in the manuscript for this method (41, 42, 106). The rationale behind the method is better described in Brady et al., *Genes & Development*, 2009, which was provided as a reference in the authors response. This reference should be added to the paper, along with a summary of how the score used relates to a typical ROC analysis.

4) The origin of the 28k insertion sites used in Figure 4A is still not clear. Presumably this is the combination of the ~14k sites defined at the very beginning of the Results with the ~14k sites identified in this study in CD4+ T cells without JQ1 treatment?

5) The authors state that they have updated the figure legends, main text, and methods to clarify which analyses are restricted to protein coding genes, however this information still appears to be missing.

6) While the relevance of the analysis in Figure 6E-F is now better explained in the manuscript, this figure does not support the authors claims that RIGs cluster in 3D space. In order to support this claim, it would be necessary to show a control analysis of clustering of non-RIGs at similar genomic separations to the RIGs shown.

7) The RIG list and location data should be provided as a plain text file or excel file rather than in pdf format, to make the data reusable.

Reviewer #3 (Remarks to the Author):

Authors present some seemingly elegant findings, most notable delineation of A1 region from Jurkat Hi-C analysis and striking correlation with HIV-1 integration frequency (though, also see below).

Some other data is much more confirmatory of earlier work, such as Fig 1A, or basically negative in nature (and confirmatory), such as Fig 3A and B. Showing negative data in main text of high impact paper is counterintuitive.

As I understand it, authors conclude certain RIG targets move from central toward peripheral region of nucleus upon T cell activation, presumably now making them preferred integration targets, in line with their prior report that RIGs preferentially map to periphery (2015 Nature). Though an intriguing observation, authors stop short of direct, critical test. If mechanistically relevant, mobile RIGs would be significantly underutilized in resting T cells. Though it is surely more challenging to infect resting as compared to activated T cells, this nevertheless is highly approachable. It is critical to obtain resting T cells from 2 to 3 blood donors, split these each into 2 pots, activate one pot each, and then infect both pots with HIV-1. Then determine integration sites and RIG usage. To date, no one has reported such significant differences in integration targeting between resting and activated T cells, yet the mobile RIG model would surely predict this.

The paper is underdeveloped in other ways. Authors correctly cite preference for HIV-1 to integrate into transcriptionally active genes, but very much under develop known mechanisms to date. LEDGF is only mentioned once in passing, yet numerous papers (uncited here) previously showed key role for LEDGF to target integration to transcriptionally active genes. It is pity authors do not investigate role of LEDGF here. There are small molecule inhibitors of LEDGF-integrase interaction commercially available, yet authors instead choose to study JQ1 and present negative results, which merely confirms earlier negative result work. Such work with LEDGF would have increased impact significantly.

Second known targeting mechanism, not discussed at all here, is capsid-CPSF6 interaction (PMID: 22174692, 23097450, 26586435, 26858452, 27307565, 30173955). Though clean small molecule inhibitors of capsid-CPSF6 interaction are not available, published capsid mutant viruses phenocopy effect of CPSF6 depletion.

Thus, relatively straightforward experiments, with LEDGIN and CA mutant virus, could have addressed known mechanisms of HIV-1 integration targeting through established virus-host interactions.

Authors discuss p300, which can bind HIV-1 integrase, as potential player. In sharp contrast to known roles for integrase-LEDGF and capsid-CPSF6, a role for p300-integrase interaction in HIV-1 integration targeting has yet to be established.

A key major concern is how authors define RIGs. Two different, though related, selection pressures define integration site patterns. During acute infection phase (circa 1 day to week or 2), integration is highly targeted to active genes basically irrespective of cell type (ref 24, PMID: 27307565; also reproduced here). This initial preference, in at least some cell types, is basically completely governed through LEDGF-integrase and capsid-CPSF6 interactions (PMID: 26858452), and seems to be main focus of present work. In sharp contrast, only small subset of such sites persist under cART selection pressure (ref 24). Thus, although sites in HIV patients are part of initial provirus population, they could (and in some cases almost certainly were) not highly targeted in initial infection phase. To group such integration site information together to define one RIG dataset, as apparently done here, should be avoided and may be seriously flawed. Such criticism was also raised by initial reviewer.

Secondary issues:

1. Page 5 line 15: “as expected”. Please include relevant citation(s).
2. Page 5, line 20: “a tropism that is not a general feature of retroviruses” is pedantic and should be avoided. Field over past 20 years has established different targeting preferences for different classes of retroviruses.
3. Page 6 bottom section “Our previously published results showed that RIGs are distributed in the periphery of the T cell nucleus”. Such findings are not reproduced in some labs (PMID: 30173955).
4. Peripheral localization of KDM2A and PACS1 in activated T cells (page 8) is not always observed (Achuthan, 2018 CHM).

Response to the Reviewers by Lucic et al.

Overall, the authors have responded in a satisfactory way to many of the suggestions in the previous review. The revised manuscript is clearer and the background and context are described in more detail, enabling readers to better appreciate the significance and relevance of the results. However, I have a few remaining concerns that should be addressed before I'd support publication of this manuscript.

We wish to thank the reviewer once again for all the comments and suggestions, which contributed significantly to improve our manuscript. Below our response to the remaining concerns (in blue)

Major concerns:

1) Thank you to the authors for including statistical modelling of the contributions of different factors to HIV integration. This definitely has the potential to clarify the results of the manuscript and they key features affecting HIV integration. However, I have some concerns about the choice of features to predict and their relation to the other analyses.

a. In the statistical modelling, the authors choose to predict "typical targets" and "hotspots". These definitions are not used elsewhere in the manuscript, which makes it difficult to interpret the modelling results in relation to the other analyses. How do these two classes relate to each other and to the previously defined RIGs?

Hotspots are genomic regions enriched in integration sites. Typical targets are genomic entities that are known to be integration spot. RIGs correspond to genes with an insertion hotspot in Jurkat cells (right). Actually, RIGs do not have a globally higher insertion rate per Mb on average when compared to expressed genes (left, expressed genes are defined as > 1 FPKM).

b. In particular, it's not clear why the authors chose to consider insertion rate normalised to gene length in order to define "typical targets" here, rather than just the total number of insertions per gene as used to define RIGs.

The proper way to measure HIV insertion rate is to normalize it by gene size (otherwise very large genes receive more HIV insertions only because of their size). This is straightforward for mapping data from a single experiment, but there is no obvious way to do it for data that consists of gene lists in different patient studies. We are conscious that this causes a bias: RIGs are on average three times larger than other genes. However, the proportion that contains a hotspot is nearly eight times higher than in expressed genes (see bar plot above), so the tropism of HIV is not explained only by the large size of the RIGs.

We wished to remove gene size as a confounding factor in the statistical modelling, therefore we used the normalized insertion rate.

c. The "hotspots" are also not sufficiently described. They appear to be defined as 10kb bins, but then they should all have the same size, so why is "size" a factor in the model?

A hotspot is a 10 kb window in the top 2.5% HIV insertion rate (corresponding to strictly more than 5 insertions). The variable "size" refers to the gene, which has an impact because long genes are more likely to have a hotspot than short genes. The reason was the same as above: we must include this variable in the model because it is a confounding factor. This has been clarified in the text.

Are the hotspots primarily found in non-coding regions, or do the authors only consider those that overlap with genes?

We considered only the hotspots in genes. The motivation for this analysis was to tease apart the contribution of gene expression versus genome organization, so we excluded intergenic regions with the rationale that they cannot be expressed. With the definitions of the manuscript, 4.8% of the hotspots are outside genes (64 out of 1,335). They are few and including them did not address the question of the Reviewer so we left them out of the analysis.

d. Most importantly, are the results of the modelling analysis robust to choosing different thresholds for the definitions of typical targets and hotspots? The most convincing analysis here would be to show that RIGs can be predicted, rather than adopting new definitions solely for this analysis.

The plots below show the predictions of the models when the thresholds are changed.

On the left we used less conservative thresholds (typical targets are defined as top 50% and hotspots are defined as genes with a 10 kb bin in the top 5% highest insertion rate), on the right we used more conservative thresholds (typical targets are defined as top 20% and hotspots are defined as genes with a 10 kb bin in the top 1% highest insertion rate). Regardless of the threshold, “typical” targets are best predicted by gene expression and the presence of a hotspot is predicted by a combination of three factors.

Predicting RIGs instead of hotspots presents two issues: the first is that RIGs are defined from patient studies but the genomic features are measured in Jurkat cells. Defining RIGs in Jurkats (as it was done on patient samples) was not possible, as we did not perform multiple and unrelated sequencing experiments with different batches of Jurkat cells. The second is that RIGs have an inflated size due to the way they are identified. As a consequence, the variable “size” masks the contribution of the other two (below) and makes the results hard to compare.

It would be more consistent with the narrative of the manuscript to train the model on RIGs, but they are not suited for this type of analysis. We have chosen the consistency of the data, with the risk that it can cause some confusion. The discrepancy above is not necessarily worrisome. RIGs have a statistical definition, so there are some false positives that overemphasize the importance of the “size” variable in this case. RIGs are still strongly enriched in hotspots as shown in the first figure above, consistently with the main messages and the main results of the manuscript.

2) Thanks also for the clarification of the Hi-C analysis pipeline that was used to produce the compartment assignments. Was the normalised Hi-C data also used for the interchromosomal interaction strength analyses presented in Figure 4? It is vital that normalised data is used here, as similar results would be produced if biases in the Hi-C procedure led to increased raw interaction strengths at SEs and/or active genes (due to accessibility, GC content, mappability, etc...). Figure 4C has low image quality, but appears to show uneven “stripes” in the Hi-C matrix which are characteristic of unnormalised Hi-C data, leading me to be concerned about the processing of the data used for the analyses shown in this figure.

The Hi-C data was normalized for all the analyses. The sample in Fig. 4 shows the unnormalized data on purpose so that the readers can see that TADs and loops are visible without preprocessing. This has been clarified in the legend. The mini panels of Fig. 5 have only a symbolic value to explain the method to identify sub-compartments (note that they have no scale nor indication of location for instance). We can replace them with more realistic samples from normalized data if the Editor recommends it.

Minor concerns:

3) The calculation of the “ROC area under curve” values is still not sufficiently explained. Typically, ROC analysis involves plotting true positive rate vs false positive rate for varying parameter sets or thresholds. The analysis as described in the methods does not appear to relate to this typical ROC

analysis, and I was unable to find any further clarification of this in the references provided in the manuscript for this method (41, 42, 106). The rationale behind the method is better described in Brady et al., *Genes & Development*, 2009, which was provided as a reference in the authors response. This reference should be added to the paper, along with a summary of how the score used relates to a typical ROC analysis.

The authors have updated the references in the methods and the main text, as well as expanded the explanation for ROC analysis: For every chromatin feature and experiment we analysed, we compared the density of values for this feature measured on integration sites, versus density of values of this feature measured on control sites. For every cut-point of the value of the measured feature, we measured the percentage of integration sites with a value of this feature higher than the cutpoint (true positive rate) and percentage of control sites with a value of this feature higher than the cut-point (false positive rate). Thus, we constructed the ROC curve by calculating the true and false positive rates for all possible cut-point values for the analysed epigenomic feature. The area under the ROC curve was then calculated.

4) The origin of the 28k insertion sites used in Figure 4A is still not clear. Presumably this is the combination of the ~14k sites defined at the very beginning of the Results with the ~14k sites identified in this study in CD4+ T cells without JQ1 treatment?

The reviewer is right, the remaining ~14k sites are identified in this study in CD4+ T cells without JQ1 treatment by inverse PCR. We have adjusted the sentence in the result section and we hope that this is more clear now.

5) The authors state that they have updated the figure legends, main text, and methods to clarify which analyses are restricted to protein coding genes, however this information still appears to be missing.

The authors apologize for this oversight. We have updated the methods (Bioinformatic analysis: Integration sites and genes) with the following sentences: In those cases where using all genes could have introduced bias to the results (for ChIP-Seq profiles on genes and expression data), only protein coding genes were used for the analysis. This was the case in the following figures: Figure 1A, Figure 2A, 2B, 2C, 2D, Figure 3D, Figure 5F, Supplementary figure 2 (B, C) and Supplementary figure 5B. List of all genes from Ensembl, GRCh37 assembly from February 2014 was used for the Figures 1C, Supplementary Figure 1 (A, B, C, D) and Supplementary Figure 3.

We have also updated the following figure legends: Figure 1A, Figure 2A, 2B, 2C, 2D, Figure 3D, Figure 5F, Supplementary figure 2 (B, C) and Supplementary figure 5B. They now clearly state that the data shown refers to protein coding genes. Finally, we have also updated the main text accordingly.

6) While the relevance of the analysis in Figure 6E-F is now better explained in the manuscript, this figure does not support the authors claims that RIGs cluster in 3D space. In order to support this claim, it would be necessary to show a control analysis of clustering of non-RIGs at similar genomic separations to the RIGs shown.

Our hypothesis about spatial clustering of HIV-1 RIGs developed from our previous observations that HIV-1 insertion sites cluster when mapped on chromosomes in certain regions of the human genome (Marini et al *Nature* 2015). This idea was nicely confirmed at the populational level by HI-C, while representative FISH labelled nuclei were used as an example of single cell HIV-1 targeted gene contacts.

HTI FISH experiments were thus designed to address the spatial proximity of certain HIV-1 RIGs, as depicted on the schemes above images in Figure 6E and 6F. As a molecular ruler of what is spatially proximal, we used internal controls from the same chromosome (chr 17) where striking differences were obtained for contacts between genes which are found on the similar linear distances. Namely, NPLOC4 – RPTOR do not cluster (measured spatially 1.91 μm) despite being separated linearly by 0.8 Mb, while GRB2 -TNRC6C from the same locus, which are separated linearly by 2.6 Mb are found on the 0.598 μm spatial distance. All these measurements are summarized in the Supplementary Figure 6F with the example images shown below.

While we agree with the reviewer that other genes, ie non-RIGs should be tested (approximately 44000 genes which are never found to have any integrations, see excel file for Additional File 2), we feel that the scope of our claim is not as broad to require such labor extensive analysis. Moreover, picking these non-RIG genes would require much more comprehensive reasoning of how to pick the candidates, rather than looking only at distances between genes. As we are showing HTI as an illustration of the globally observed phenomenon, we believe that internal molecular ruler could support our conclusions that a portion of HIV-1 targeted genes display clustering in the 3D space.

7) The RIG list and location data should be provided as a plain text file or excel file rather than in pdf format, to make the data reusable.

We have now provided the excel file with all genes that were included with analysis along with the number of experiments that found that gene to have an integration (Additional file 2).

Reviewer #3 (Remarks to the Author):

Authors present some seemingly elegant findings, most notable delineation of A1 region from Jurkat Hi-C analysis and striking correlation with HIV-1 integration frequency (though, also see below).

We wish to thank the Reviewer for considering the main conclusions of our work elegant, and for valuable comments and remarks.

We think that the revised version of our manuscript has strongly benefited from the Reviewer's suggestions. Below, our responses are in blue.

Some other data is much more confirmatory of earlier work, such as Fig 1A,

We appreciate Reviewer's remarks, and we agree that the data in **Figure 1A** are in line with previous HIV-1 integration studies, as indeed ChIP-Seq profiles confirm that genes targeted by HIV-1 are enriched with open chromatin marks, such as H3K36me3 and H3K4me1, while depleted of H3K27me3^{1,2}. We used this data as both positive and negative markers of HIV-1 integration to develop on striking H3K27ac enrichment, and associated BRD4 and MED1 which are SE constituents, one of the new findings of here presented work. It had escaped our attention that the text presents these findings as novel. We have edited the text to make this point clear.

or basically negative in nature (and confirmatory), such as Fig 3A and B. Showing negative data in main text of high impact paper is counterintuitive.

We are thankful to the Reviewer for raising concerns about the clarity and flow of the main text. We believe that these results are central to the present investigation, but we agree that they may not be emphasised. We have opted to add the set of data presented in the main text in **Figure 3A** and **3B** (integration sites repertoire in JQ1 treatment) with the accompanying FISH and RNA-Seq data (complete **Figure 3**) to the Supplementary **Figure 2**. Although the main message based on our integration site sequencing and RNA-Seq expression data that HIV-1 maintains integrations into highly transcribed genes is important, it is not novel, and can be conveyed as a supporting file.

As I understand it, authors conclude certain RIG targets move from central toward peripheral region of nucleus upon T cell activation, presumably now making them preferred integration targets, in line with their prior report that RIGs preferentially map to periphery (2015 Nature). Though an intriguing observation, authors stop short of direct, critical test. If mechanistically relevant, mobile RIGs would be significantly underutilized in resting T cells. Though it is surely more challenging to infect resting as compared to activated T cells, this nevertheless is highly approachable. It is critical to obtain resting T cells from 2 to 3 blood donors, split these each into 2 pots, activate one pot each, and then infect both pots with HIV-1. Then determine integration sites and RIG usage. To date, no one has reported such significant differences in integration targeting between resting and activated T cells, yet the mobile RIG model would surely predict this.

We appreciate the valuable comments regarding HIV-1 targets in the resting CD4⁺ T cells population, and we agree with the Reviewer that it is important to investigate HIV-1 integration patterns in this cell state.

However, due to multiple restrictions taking place before the integration of the provirus, the infection rate in resting cells is insufficient to carry out this experiment³⁻⁶. Specifically, we

obtained only 48 sites from donor 1R and 19 sites from donor 2R (corresponding insertion sites from activated counterparts are 1907 (1A) and 612 (2A), respectively).

Such a small number of integration sites hampers the statistical analysis. Fisher's exact test shows no significant difference between resting and activated CD4⁺ T cells. In resting cells, 44 genes are targeted with 67 insertions, versus 1218 genes targeted from 2519 insertions in activated CD4⁺ T cells. Out of those, 15 from resting cells (24%) and 343 genes from activated cells (28%) are classified as RIGs (as defined in activated cells). As mentioned, this difference is not statistically significant (one-sided Fisher's exact test, odds ratio 1.32, p-value 0.84, the hypothesis of interest is that HIV is found *less often* in RIGs in resting cells).

However, in resting cells those genes are more likely to be targeted by pure chance due to their length: out of 15 RIGs targeted in resting cells, 14 are in the top 10% by gene length (93%), while this is true for 242 out of 343 RIGs targeted in activated cells (70%) Figure R1. We found a significantly higher fraction of long genes (top 10% of gene length) among HIV-1 targets in resting cells than in activated cells (one-sided Fisher's exact test, odds ratio 5.82, p-value 0.042, the hypothesis of interest is that HIV is found *more often* in long genes in resting cells).

In summary, our analysis shows that there is no statistically significant difference between RIGs utilization in resting and activated CD4⁺ T cells, however, RIGs targeted in resting T cells are significantly longer than RIGs targeted in activated cells.

Figure R1:

Fraction of RIGs targeted in activated and resting cells that are among 10% longest genes. The proportion of long genes among targeted genes in resting cells is significantly higher than the proportion of long genes which are targeted in activated cells is statistically significant, (one-sided Fisher's exact test for count data, odds ratio=5.83 (95% conf. interval 1.050456 - Inf), p-value 0.042.

The paper is underdeveloped in other ways. Authors correctly cite preference for HIV-1 to integrate into transcriptionally active genes, but very much under develop known mechanisms to date. LEDGF is only mentioned once in passing, yet numerous papers (uncited here) previously showed key role for LEDGF to target integration to transcriptionally active genes. It is pity authors do not investigate role of LEDGF here. There are small molecule inhibitors of LEDGF-integrase interaction commercially available, yet authors instead choose to study JQ1 and present negative results, which merely confirms earlier negative result work. Such work with LEDGF would have increased impact significantly.

Second known targeting mechanism, not discussed at all here, is capsid-CPSF6 interaction (PMID: 22174692, 23097450, 26586435, 26858452, 27307565, 30173955). Though clean small molecule inhibitors of capsid-CPSF6 interaction are not available, published capsid mutant viruses phenocopy effect of CPSF6 depletion.

Thus, relatively straightforward experiments, with LEDGIN and CA mutant virus, could have addressed known mechanisms of HIV-1 integration targeting through established virus-host interactions.

Authors discuss p300, which can bind HIV-1 integrase, as potential player. In sharp contrast to known roles for integrase-LEDGF and capsid-CPSF6, a role for p300-integrase interaction in HIV-1 integration targeting has yet to be established.

We agree with the Reviewer that viral proteins integrase and capsid and their main interactors LEDGF/p75 and CPSF6 are the principal determinants of HIV-1 integration targeting. We are aware of the numerous studies which showed a key role for LEDGF/p75 in targeting HIV-1 integration to the bodies of transcriptionally active genes ⁷⁻¹². Small molecule inhibitors of LEDGF/p75 - integrase interaction, LEDGINs are indeed interesting tools. They efficiently retarget integration patterns out of transcription units ¹¹ in such a way that the authors propose to use them to “block and lock” the provirus in transcriptionally dormant, latent state ¹³. We agree that this would be interesting to further explore the integration patterns in the presence of LEDGINs, but this is out the scope of our manuscript.

CA-CPSF6 interaction is also very important in guiding HIV-1 preintegration complexes to the regions of the nucleus where then HIV-1 IN through its interaction with LEDGF inserts the viral genome into cellular chromatin ¹⁴. There are indeed different CA mutants that can be used for the purposes of retargeting, like P90A, N74D ^{15,16} or V77A ^{17,18}. However, such retargeting would lead us to the regions of chromatin that are normally avoided by the wild type HIV-1; for example V77A mutant targets HIV-1 integrations to Lamina Associated Domains, which are excluded by normal HIV-1 integrations. Similarly to what explained above, although it would be interesting and important to explore CA mutants in T lymphocytes, this will be part of future studies.

We appreciate the comment the reviewer made, and have now added a paragraph to the introduction and have also discussed our data in the light of some of the above mentioned works on LEDGF-IN inhibitors and CA mutants.

A key major concern is how authors define RIGs. Two different, though related, selection pressures define integration site patterns. During acute infection phase (circa 1 day to week or 2), integration is highly targeted to active genes basically irrespective of cell type (ref 24, PMID: 27307565; also reproduced here). This initial preference, in at least some cell types, is basically completely governed through LEDGF-integrase and capsid-CPSF6 interactions (PMID: 26858452), and seems to be main focus of present work. In sharp contrast, only small subset of such sites persist under cART selection pressure (ref 24). Thus, although sites in HIV patients are part of initial provirus population, they could (and in some cases almost certainly were) not highly targeted in initial infection phase. To group such integration site information together to

define one RIG dataset, as apparently done here, should be avoided and may be seriously flawed. Such criticism was also raised by initial reviewer.

We are grateful to the Reviewer for this remark. The results from the cited literature are indeed very clear, but they pertain to the tropism of HIV towards active genes, which is not the same as the tropism towards RIGs, as we show in the manuscript. The argument of the Reviewer is sound, but it is based on indirect evidence. We bring direct evidence in the manuscript and in the answer below that the high enrichment of HIV in RIGs is similar in patients and in *in vitro* systems. We insist that this is not incompatible with the cited literature, as the authors did not test the tropism of mutant HIV relative to RIGs or insertion hotspots.

Consistent with the previous studies¹⁹, we find that 80% of integrations are inside genes, in both *in vitro* infections and in patient data. We use recurrent integration genes (RIGs) as genomic determinants of HIV-1 integration. A gene is a RIG if it contains at least a single integration and is found in at least 2 different studies (ie labs, sequencing methods, *in vitro* or patient data sets). We do not account for multiple integrations per gene, nor clonally expanded integration sites (limited remaining integrations selected on ART).

Similar definition of RIGs is given in the main text, and we hope this is sufficiently clear: 'We thus defined Recurrent Integration Genes (RIGs) as genes with at least a single HIV-1 integration in at least two out of eight datasets, yielding a total of 1,831 RIGs'

We share the Reviewer's concern that two different selection pressures define integration site patterns. Indeed it is something we were aware of when comparing patient data with integrations from *in vitro* primary CD4⁺ T cell infections. However, our analysis showed that the variance between measured features in different *in vitro* / patient derived data sets is higher than the one between *in vitro* and patient data (**Figure 1B** and **Figure R2**). Similarly, when observing targeted genes in the experiments, we also conclude that the differences between different data sets are higher than are differences between patients and *in vitro* data sets.

In particular, we compared patient data with integration sites in primary infection, defined the chromatin profiles and then analyzed chromatin mark distributions on the integration sites separately for each data set (study). In **Figure 1B** and **Figure R3** we show co-occurrence of integration sites and epigenetic modifications obtained by ChIP-Seq for H3K27ac, H3K4me1, BRD4, MED1, H3K36me3, H4K20me1, H3K4me3, H3K27me3 and H3K9me2, for all data sets separately. Columns marked as Brady and Lucic correspond to *in vitro* experiments while the other columns show results from patients divided in different lists. In **Figure R2**, columns marked Patients and InVitro show the results from all integration sites from patient data sets and *in vitro* data sets, respectively. The column marked TotalCD4HIV shows the results for all integrations from all HIV-1 datasets we used, while columns marked HTLV and MLV show results from data sets corresponding to HTLV and MLV integration sites. Not surprisingly, when observing average linkage hierarchical clustering results shown in the **Figure R2** above the columns, we can see that all HIV data sets group together, separately from HTLV and MLV data sets. Moreover, there are 2 distinct clusters within the HIV-1 datasets. The first one contains Han, Kok, Ikeda, Brady, Maldarelli and Wagner data sets, while the second cluster groups Lucic, InVitro, Cohn, Patients and totalCD4HIV datasets of integration sites. Thus, we do not observe

separation of clusters of patient-derived data vs. data sets obtained *in vitro* based on the ROC analysis of values of epigenetic features surrounding the integration sites.

Figure R2: Results of the ROC analysis represented as a heatmap summarizing the co-occurrence of integration sites and epigenetic modifications obtained by ChIP-Seq for BRD4, H3K27ac, H3K27me3, H3K36me3, H3K4me1, H3K4me3, H3K9me2, H4K20me1 and MED1. HIV-1, HTLV and MLV integration data sets are shown in the columns, and epigenetic modifications are shown in rows. Associations are quantified using the ROC area method; values of ROC areas are shown in the color key at the right. Columns representing data sets are clustered by using average linkage hierarchical clustering of the calculated values.

When integration preferences are analysed at the level of genes, again no separation of patient vs *in vitro* data sets can be observed. This conclusion is reached when looking at the percentage of overlaps between all pairs of data sets. The following heatmap (**Figure R3**) shows hierarchical clustering of data sets based on their pairwise similarities. Each cell represents the fraction of genes found in data set X (rows) that is also found in data set Y (columns). Note that this matrix is not diagonally symmetric due to different sizes of the data sets (% of A in B is different from % of B in A). When applying hierarchical average linkage clustering, we observe that smallestRandom and largestRandom mock data sets cluster together as expected and that there is no separation of patient and *in vitro* (Brady and Lucic)

data sets. We decided to show supplementary figure 3 instead of this figure due to simplicity. Supplementary figure 3 shows the fraction of genes from each data set that is shared with at least one other data set. Additionally, we have calculated distances from the genes targeted by *in vitro* data sets and patient derived integration sites to the nearest super-enhancer. No significant differences are found, as shown on **Figure R4**. We conclude that there is no difference in integration site preferences relative to the features of interest in this study in data sets obtained from patients and the ones obtained from *in vitro* experiments.

Figure R3: Heatmap of pairwise similarities between data sets. Each cell represents fraction of unique genes overlapping integrations from data set in the column with unique genes overlapping integrations in data set in the row. Data sets are clustered using hierarchical clustering with average linkage as a method to calculate distances between clusters. Data sets marked as Brady and Lucic correspond to *in vitro* experiments while the rest of the data sets originate from patients. smallestRandom and largestRandom represent control (mock) data sets.

Finally, and related also to the previous comment of the Reviewer, we wish to clarify and we have now highlighted in the manuscript that in most cell types, integration into the human genome, including integration into RIGs is mediated by the viral factor IN and its interacting partner LEDGF/p75, while selection of the chromatin environment is highly depends on viral CA and CPSF6 cellular factor. Although the focus of our work, was to complement the current knowledge on HIV-1 integration sites from the perspective of their chromatin, genomic and 3D features, we have now modified the text to underline that we are not revisiting the well grounded integration mechanisms, but rather we bring new information on the relationship between the 3D conformation of the genome and the HIV-1 integration landscape.

Figure R4: Distances to nearest super-enhancer for genes with integration sites from patient and *In Vitro* data sets.

Secondary issues:

1. Page 5 line 15: “as expected”. Please include relevant citation(s).

We thank the Reviewer for his comment, and we now include in the manuscript the missing citation: ²⁰.

2. Page 5, line 20: “a tropism that is not a general feature of retroviruses” is pedantic and should be avoided. Field over past 20 years has established different targeting preferences for different classes of retroviruses.

We agree with the Reviewer with his remark and we have now changed this statement in the current revised version of the

manuscript.

‘Thus, HIV-1 tends to integrate into genes proximal to super-enhancers, a tropism that is not a general feature of retroviruses’ will be exchanged with ‘Thus, HIV-1 displays specific preference to integrate into genes proximal to super-enhancers, newly defined genomic elements of retroviral integrations.’

3. Page 6 bottom section “Our previously published results showed that RIGs are distributed in the periphery of the T cell nucleus”. Such findings are not reproduced in some labs (PMID: 30173955).

4. Peripheral localization of KDM2A and PACS1 in activated T cells (page 8) is not always observed (Achuthan, 2018 CHM).

(This answer refers to points 3&4)

We are thankful to the Reviewer for these critical comments. While our previous study ² and the data presented in this work argue for a more peripheral localization of HIV-1 recurrent integration genes in the nucleus of primary activated CD4 T cell, we agree that more robust analysis (which is the aim of our follow up), such as High Throughput Imaging mapping of the sub-compartments (in particular A1) should be conducted to obtain comprehensive results on positioning of all HIV-1 target genes.

We also agree with the Reviewer that in particular chr 11 RIGs, KDM2A and PACS1, do not display peripheral distribution. We are including here the FISH analysis by nuclear zones as defined previously by us and others. The radial positioning of both regions (genes) changes upon T cell activation, from internally distributed zones 3 and 2 to the more peripheral regions of zone 2 and 1.

Figure R5 Three-dimensional immuno-DNA FISH of KDM2A and PACS1 RIGs in resting and activated (anti-CD3/anti-CD28 beads, IL-2 for 48 h) CD4⁺ T cells (green: BAC/gene probe, red: lamin B1, blue: DNA staining with Hoechst 33342, scale bar represents 2 μ m). Bar plots represent percentages of alleles in nuclear zones, where 1 is peripheral, and 3 central. Signals with signal-to-radius ratios smaller than 0.19 were assigned to zone 1, signals with ratios ranging from 0.19 to 0.43 to zone 2, and signals with ratios higher than 0.43 to zone 3 (as in ²). Cumulative frequency plots show combined data from both experiments (n = 100, black: resting cells, red: activated cells). The p-values of the Kolmogorov-Smirnov tests are indicated.

We have now changed the text in the revised manuscript, 'RIGs are distributed in the periphery of the T cell nucleus' with 'majority of tested RIGs are distributed in the outer zones of the T cell nucleus'.

1. Brady, T. *et al.* HIV integration site distributions in resting and activated CD4⁺ T cells infected in culture. *AIDS* **23**, 1461–1471 (2009).
2. Marini, B. *et al.* Nuclear architecture dictates HIV-1 integration site selection. *Nature* **521**,

227–231 (2015).

3. Manganaro, L. *et al.* Concerted action of cellular JNK and Pin1 restricts HIV-1 genome integration to activated CD4+ T lymphocytes. *Nat. Med.* **16**, 329–333 (2010).
4. Baldauf, H.-M. *et al.* SAMHD1 restricts HIV-1 infection in resting CD4(+) T cells. *Nat. Med.* **18**, 1682–1687 (2012).
5. Sheehy, A. M., Gaddis, N. C., Choi, J. D. & Malim, M. H. Isolation of a human gene that inhibits HIV-1 infection and is suppressed by the viral Vif protein. *Nature* **418**, 646–650 (2002).
6. Sheehy, A. M., Gaddis, N. C. & Malim, M. H. The antiretroviral enzyme APOBEC3G is degraded by the proteasome in response to HIV-1 Vif. *Nat. Med.* **9**, 1404–1407 (2003).
7. Ciuffi, A. *et al.* A role for LEDGF/p75 in targeting HIV DNA integration. *Nat. Med.* **11**, 1287–1289 (2005).
8. Shun, M.-C. *et al.* LEDGF/p75 functions downstream from preintegration complex formation to effect gene-specific HIV-1 integration. *Genes Dev.* **21**, 1767–1778 (2007).
9. Schrijvers, R. *et al.* HRP-2 determines HIV-1 integration site selection in LEDGF/p75 depleted cells. *Retrovirology* **9**, 84 (2012).
10. Ferris, A. L. *et al.* Lens epithelium-derived growth factor fusion proteins redirect HIV-1 DNA integration. *Proc. Natl. Acad. Sci. U. S. A.* **107**, 3135–3140 (2010).
11. Vranckx, L. S. *et al.* LEDGIN-mediated Inhibition of Integrase-LEDGF/p75 Interaction Reduces Reactivation of Residual Latent HIV. *EBioMedicine* **8**, 248–264 (2016).
12. Marshall, H. M. *et al.* Role of PSIP1/LEDGF/p75 in Lentiviral Infectivity and Integration Targeting. *PLoS One* **2**, e1340 (2007).
13. Debyser, Z., Vansant, G., Bruggemans, A., Janssens, J. & Christ, F. Insight in HIV Integration Site Selection Provides a Block-and-Lock Strategy for a Functional Cure of HIV Infection. *Viruses* **11**, (2018).
14. Sowd, G. A. *et al.* A critical role for alternative polyadenylation factor CPSF6 in targeting HIV-1 integration to transcriptionally active chromatin. *Proc. Natl. Acad. Sci. U. S. A.* **113**, E1054–63 (2016).
15. Schaller, T. *et al.* HIV-1 capsid-cyclophilin interactions determine nuclear import pathway, integration targeting and replication efficiency. *PLoS Pathog.* **7**, e1002439 (2011).

16. Koh, Y. *et al.* Differential effects of human immunodeficiency virus type 1 capsid and cellular factors nucleoporin 153 and LEDGF/p75 on the efficiency and specificity of viral DNA integration. *J. Virol.* **87**, 648–658 (2013).
17. Achuthan, V. *et al.* Capsid-CPSF6 Interaction Licenses Nuclear HIV-1 Trafficking to Sites of Viral DNA Integration. *Cell Host Microbe* **24**, 392–404.e8 (2018).
18. Saito, A. *et al.* Capsid-CPSF6 Interaction Is Dispensable for HIV-1 Replication in Primary Cells but Is Selected during Virus Passage In Vivo. *J. Virol.* **90**, 6918–6935 (2016).
19. Schröder, A. R. W. *et al.* HIV-1 integration in the human genome favors active genes and local hotspots. *Cell* **110**, 521–529 (2002).
20. LaFave, M. C. *et al.* MLV integration site selection is driven by strong enhancers and active promoters. *Nucleic Acids Res.* **42**, 4257–4269 (2014).

Reviewers' comments:

Reviewer #1 (Remarks to the Author):

The descriptions of the ROC analysis, features used for statistical modelling, and Hi-C processing are now much clearer. Thank you to the authors for making these changes! In addition, it's good to see that the modelling results are robust to different thresholds. I'm happy with the authors' explanations of the choice of features to predict for modelling and the inclusion of gene size in the model.

I'm also satisfied that the claims made in relation to Figure 6 E and F are supported by the data and don't overstate the significance of these results from a limited number of loci.

The current Figure 4C doesn't show any artefacts in the Hi-C data (probably these were due to low image quality in the previous manuscript version). Overall, I'm satisfied with the changes made by the authors and would now support publication of the manuscript.

Reviewer #3 (Remarks to the Author):

Paper has improved, but some issues remain. For one, despite multiple attempts, I was unable on reviewer portal to locate Supplementary Figures. Thus, I was unable to fully evaluate the revised paper. If the figures were there and just my stupidity precluded me from finding them, I wholeheartedly apologize.

The strength of the paper lies with the identification of sub-Jurkat cell compartments by HI-C and their correlation to HIV integration targeting. Though deciphering the contributions of LEDGF and CPSF6, whose known roles in integration targeting are now documented, would surely increase the impact of the work, I can agree with the authors that such work exceeds the scope of current paper.

Remaining major issues:

1. The RIG definition method is seemingly flawed because it fails to account for extent of integration targeting expected from random chance. Most if not all integration mapping studies run parallel in silico analyses to match their wet bench pipeline for DNA preparation to define what would occur by chance in absence of virus infection. In other words, if your genomic DNA is cut with restriction enzymes A and B for your LM-PCR or inverse PCR strategy, then human genome build 38 is processed same way in silico to define all possible A-A, A-B, and B-B fragments, which are then used to define random frequency for any genomic annotation you map.

Problem with approach herein is that authors more simply define RIGs as genes that were targeted for integration in at least 2 of 7 studies, which is much more arbitrary than tried and true approach of precluding genes based on statistically defined random cut-off (even if they showed in 2 of 7 studies). Others who have used statistics-based approach (ref 20) find that HIV-1 RIGs are relatively small, which does not seem to be case here and calls into question the utilized procedure.

2. Authors seem to agree that RIGs in association with SEs that are seen to move peripherally upon T cell activation should be preferred targets in activated versus quiescent T cells. In rebuttal letter they analyzed total of 67 sites from resting cells and 2519 from activated cells, concluding that low number of resting cell sites confounded the analysis due to lack of apparent statistical power (though, an associated power analysis was not performed).

This is fairly disingenuous response. Authors heavily rely on integration sites from ref 44, which

included 3031 sites from quiescent cells and 2683 sites from activated cells. This surely should be sufficient sites to perform the requested analyses. It is critical to apply the RIG analysis and report the results in the paper.

3. The DNA sequences that define the different Jurkat cell compartments (A1, A2, AB, B1, B2; Fig 4C) will be extremely useful to the community, and should be made public upon acceptance of the paper.

Secondary issues:

4. Though authors now cite ref 20, they avoid key conclusion of this paper that peripheral region of the nucleus is not preferentially targeted for integration by HIV-1. Taking big picture view, while there is consensus in the field for roles of LEDGF and CPSF6 in integration targeting, role for nuclear periphery is less established. Authors should amend Abstract lines 1 and 2 and lines 7 and 8 to more accurately reflect current field status. Also page 5, new section lines 1 and 2.

5. I did locate 182 page supplemental document, which seemed just to be tables. Though even this was not so clear, as only one table, Table S3, had title, and table legends were missing completely.

6. Please consider revised Title: "Spatially clustered loci with multiple enhances are frequent targets of HIV-1 integration"

7. Intro second paragraph line 4, authors should cite original papers that established roles for NPC proteins in integration targeting PMID: 21423673, 23097450, 23523133.

8. Page 3 lower paragraph, it is entirely unclear how the non-RIG dataset can greatly exceed the number of genes in the human genome.

9. Page 4 lines 2 and 3: "We used control sites..."

10. Page 5 lower section: "we compared a published collection of 58,240 insertion sites in Jurkat cells (22) to the 28,419 insertion sites in primary CD4+ T cells from this (obtained by LAM and inverse PCR) and previous studies". The meaning of "this" is unclear: do you mean the current study, or ref 22? Also, please add citation(s) after "studies".

11. Page 10 line 11 is mangled. Please rewrite: "p300 promotes IN DNA binding activity (70) and could serve to direct viral integration toward genes with super-enhancers in the A1 compartment, though a role for p300 in HIV-1 integration targeting has yet to be established"

NCOMMS-18-14402B

Reviewers' comments:

Reviewer #1 (Remarks to the Author):

The descriptions of the ROC analysis, features used for statistical modelling, and Hi-C processing are now much clearer. Thank you to the authors for making these changes! In addition, it's good to see that the modelling results are robust to different thresholds. I'm happy with the authors' explanations of the choice of features to predict for modelling and the inclusion of gene size in the model.

I'm also satisfied that the claims made in relation to Figure 6 E and F are supported by the data and don't overstate the significance of these results from a limited number of loci.

The current Figure 4C doesn't show any artefacts in the Hi-C data (probably these were due to low image quality in the previous manuscript version). Overall, I'm satisfied with the changes made by the authors and would now support publication of the manuscript.

We are extremely grateful to this Reviewer, who was guided us through this revision process towards a much improved manuscript.

Reviewer #3 (Remarks to the Author):

Paper has improved, but some issues remain. For one, despite multiple attempts, I was unable on reviewer portal to locate Supplementary Figures. Thus, I was unable to fully evaluate the revised paper. If the figures were there and just my stupidity precluded me from finding them, I wholeheartedly apologize.

The strength of the paper lies with the identification of sub-Jurkat cell compartments by HI-C and their correlation to HIV integration targeting. Though deciphering the contributions of LEDGF and CPSF6, whose known roles in integration targeting are now documented, would surely increase the impact of the work, I can agree with the authors that such work exceeds the scope of current paper.

Remaining major issues:

1. The RIG definition method is seemingly flawed because it fails to account for extent of integration targeting expected from random chance. Most if not all integration mapping studies run parallel in silico analyses to match their wet bench pipeline for DNA preparation to define what would occur by chance in absence of virus infection. In other words, if your genomic DNA is cut with restriction enzymes A and B for your LM-PCR or inverse PCR strategy, then human genome build 38 is processed same way in silico to define all possible A-A, A-B, and B-B fragments, which are then used to define random frequency for any genomic annotation you map.

Problem with approach herein is that authors more simply define RIGs as genes that were targeted for integration in at least 2 of 7 studies, which is much more arbitrary than tried and true approach of precluding genes based on statistically defined random cut-off (even if they showed in 2 of 7

studies). Others who have used statistics-based approach (ref 20) find that HIV-1 RIGs are relatively small, which does not seem to be case here and calls into question the utilized procedure.

In order to assess confidence of RIG annotations, we generated 100 random data sets for each of the 8 mapping experiments, with equivalent number of randomly chosen integration sites. Random integration sites were constrained to match the distance to the nearest expressed gene, as explained in the ROC analysis. For each randomly generated data set, we performed the same calculation as with the real data set - every gene was assigned a value between 0 and 8 corresponding to how many times it was targeted as the (in this case random) integration site data set. By repeating this calculation 100 times, we were able to establish an “expected integration frequency” value for each gene and by comparing this value with the value observed in the real data set, we were able to assign the confidence score of each RIG. Higher scores were assigned to more confident RIGs. We have added the column with assessment of confidence to each RIG in Additional File 2 (last column nRealBetter) as well as the respective explanation of the procedure to the methods section.

2. Authors seem to agree that RIGs in association with SEs that are seen to move peripherally upon T cell activation should be preferred targets in activated versus quiescent T cells.

In rebuttal letter they analyzed total of 67 sites from resting cells and 2519 from activated cells, concluding that low number of resting cell sites confounded the analysis due to lack of apparent statistical power (though, an associated power analysis was not performed).

This is fairly disingenuous response. Authors heavily rely on integration sites from ref 44, which included 3031 sites from quiescent cells and 2683 sites from activated cells. This surely should be sufficient sites to perform the requested analyses. It is critical to apply the RIG analysis and report the results in the paper.

We agree that infection and especially integration of resting T cells is a topic of great relevance, which we partially tried to address by performing the experiment suggested by the Reviewer. Unfortunately, we obtained a very limited number of integrations, and the Reviewer rightfully suggested using the Brady study, where multiple integrations from both activated and resting CD4+ T cells were reported. The activated T cells data, obtained through the previously available link on the Bushman’s laboratory website, were already part of our manuscript and RIGs analysis, and therefore this study seemed like a reasonable and easy solution. However, their data were not available online and were deposited for both resting and activated T cells only now upon our request.

We repeated the meta analysis using their data and from 3031 raw sequences originating from resting cells, using the explained procedure from the Brady paper (removing barcodes, LTR, adapters + BLAT 98% identity), and we obtained 1441 integration sites which could be uniquely mapped to hg19 in resting cells. Some of those uniquely mappable integrations originate from clonally expanded sites, so taking this into account, we collapsed them into 1091 unique integration site in resting cells. The code for processing raw reads and getting integration sites is available at

https://github.com/gui11aume/genome_structure_and_HIV_integration/blob/master/maja/Brady_Integration_Sites.md.

In total, we found 774 genes targeted by those integration sites in resting cells, as compared to 660 in activated cells.

We compared RIG utilization in resting and activated T cells. To do this, we excluded the Brady dataset and redefines RIGs without them, as genes targeted in 2 or more data sets. Our analysis of resting and activated CD4⁺ data sets showed no statistical differences between RIGs utilization in the two cell states (proportion of targeted RIGs in activated cells=0.444; proportion of RIGs among targeted genes in resting cells=0.413; $X^2=1.2325$, p-value=0.2669).

Based on our analysis we concluded that there are no statistically significant differences between RIGs utilization in resting and activated CD4⁺ T cells.

Prompted by this lack of significant differences, and especially by the high number of integrations they reported in resting T cells, we examined the experimental conditions under which the cells were purified and Integration Site (IS) data generated. We noticed that although resting cells were isolated from the total population by addition of antibodies for T cell activation markers (negative selection), the isolated population of resting cells contained 3% of activated cells, as reported by fluorimetry (figure 1.b). The presence of this small amount of activated cells suggests that they might amplify during the experimental procedure more rapidly than the resting ones. This is especially relevant because resting cells (with 3% of activated cells in it) were kept in culture longer than the activated ones post infection (60 h vs. 36 h). To our knowledge and experience, this discrepancy between times in culture should be avoided as the number of activated cells in the resting T cell culture could increase even more. Moreover, the kinetics of the course of infection (figure 1.c) shows that resting cells are less efficient in being infected, and extended exponential infection phase suggests that active cell contaminants might play a major role in provirus/cell rate (these active cells could have internalized and integrated the virus). In addition, the multiplicity of infection (MOI) used in their study must have been very high in resting cells, as the provirus/cell ratio at 60 hours exceeds 1.

While our experimental setup generated (detected) very few integration sites in resting CD4⁺ T cells (67 only), Brady et al. found many more (up to 1087). Such high numbers go against our current understanding of resting T cell infection (Zack et al. 1990; Stevenson et al. 1990; Pierson et al. 2002; Plesa et al. 2007; Manganaro et al. 2010; Baldauf et al. 2012; Descours et al. 2012). The study by Brady et al. remains to our knowledge the only one that generated such high numbers of integrations in resting T cells.

We would like to express our concern that adding these data to our manuscript would not contribute to the overall understanding of integration sites in activated CD4 T cells. Due to the technical concerns explained above and taking into account the multiple known restrictions that limit infection of resting T cells, we are reluctant to add these results to our current manuscript.

However, if the Reviewer and the Editors after considering our points still retain that the meta analysis of RIGs behaviour in resting T cells should be added to our manuscript we will add the analysis into the additional files and comment on it discussion.

3. The DNA sequences that define the different Jurkat cell compartments (A1, A2, AB, B1, B2; Fig 4C) will be extremely useful to the community, and should be made public upon acceptance of the paper.

The documents were included in the Docker container to reproduce our analyses, but we gladly admit that this information should also be provided as a standalone file. We have added the document as a supplementary data file (in text format) and we refer to it in the methods section.

Secondary issues:

4. Though authors now cite ref 20, they avoid key conclusion of this paper that peripheral region of the nucleus is not preferentially targeted for integration by HIV-1. Taking big picture view, while there is consensus in the field for roles of LEDGF and CPSF6 in integration targeting, role for nuclear periphery is less established. Authors should amend Abstract lines 1 and 2 and lines 7 and 8 to more accurately reflect current field status. Also page 5, new section lines 1 and 2.

We wish to thank the Reviewer for her/his comment. Indeed we acknowledged the consensus role of LEDGF and CPSF6 in integration targeting. As shown in our previous (Marini et al. 2015) and current study (Figure 5) and in agreement with the viral localization in the 3D nuclear space seen by others (Lelek et al. 2015; Di Primio et al. 2013; Quercioli et al. 2016; Vranckx et al. 2016), the outer shell of the nucleus, 1 micron under the nuclear envelope in activated CD4⁺ T cell seems to be the preferred nuclear environment. A slight discrepancy with the results reported in reference 20 exists, possibly due to the activation state of CD4⁺ T cells, but the Reviewer is right, we need to accurately represent all the current literature, therefore we inserted the changes in the text as indicated.

5. I did locate 182 page supplemental document, which seemed just to be tables. Though even this was not so clear, as only one table, Table S3, had title, and table legends were missing completely.

We thank the reviewer and we have now added the missing information.

6. Please consider revised Title: "Spatially clustered loci with multiple enhances are frequent targets of HIV-1 integration"

We agree with the reviewer, and we would like to change the title, but we would also like to invite the Editor to consult us on this.

7. Intro second paragraph line 4, authors should cite original papers that established roles for NPC proteins in integration targeting PMID: 21423673, 23097450, 23523133.

We have now also included selected works that established the role of nucleoporins in HIV-1 integration.

8. Page 3 lower paragraph, it is entirely unclear how the non-RIG dataset can greatly exceed the number of genes in the human genome.

We have used the list of all genes from Ensembl, GRCh37. Beside protein coding genes, the list also includes pseudogenes, lincRNA, antisense, miRNA, and other gene biotypes. We have explained this in the methods: Gene coordinates were downloaded from Ensembl, GRCh37, February 2014. ... In those cases where usage of all (and not only coding) genes could have introduced bias to the results (for ChIP-Seq profiles on genes and expression data), only protein coding genes were used for the analysis. This was the case in the following figures: Figure 1A, Figure 2A, 2B, 2C, 2D, Figure 4F, Supplementary Figure 2 (E, G) and Supplementary Figure 5B. List of all genes from Ensembl, GRCh37 assembly from February 2014 was used for the Figures 1C, Supplementary Figure 1 (A, B, C, D) and Supplementary Figure 3.

9. Page 4 lines 2 and 3: "We used control sites..."

This line has now been corrected.

10. Page 5 lower section: "we compared a published collection of 58,240 insertion sites in Jurkat cells (22) to the 28,419 insertion sites in primary CD4+ T cells from this (obtained by LAM and inverse PCR) and previous studies". The meaning of "this" is unclear: do you mean the current study, or ref 22? Also, please add citation(s) after "studies".

We have now clarified this in the main text and added the references.

11. Page 10 line 11 is mangled. Please rewrite: "p300 promotes IN DNA binding activity (70) and could serve to direct viral integration toward genes with super-enhancers in the A1 compartment, though a role for p300 in HIV-1 integration targeting has yet to be established"

We thank the Reviewer for this comment, we have modified the main text.

REVIEWERS' COMMENTS:

Reviewer #3 (Remarks to the Author):

The paper has improved further with revision. The preferential mapping of integration sites to the A1 compartment of Jurkat T cells as assessed via novel Hi-C data is striking and carries the day. It will be interesting to see how these findings align with the works of others who do similar types of HIV integration site mappings. In this regard we are grateful the authors agree to release the sequences that define A1, A2, AB, B1 and B2 Jurkat cell regions.

Personally, I am only partially swayed by the explanation provided in the rebuttal letter as to why careful analysis of the resting versus activated integration site data provided in reference 47 failed to show different RIG usage among the datasets. The authors are concerned that 3% of the resting cell population stains for markers of cell activation. At the same time, those authors highlighted that the levels of CD25, CD69, and HLA-DR on these 3% cells are "very low" and moreover that the resting cells did not detectably proliferate as assessed by BrdU incorporation, a gold standard in the field. I therefore feel we are getting into a debate of what truly are resting CD4 T cells and whose work is to be trusted in the field. Bushman and O'Doherty are generally well regarded and document integration as carefully as anyone; their follow up paper in PLoS P (PMID 22911005) carefully plots integration over time, where the resting cell level approaches within tenfold of the activated level by day 3. This paper also reported integration sites, so it may again be a source of material for the authors to query.

However I would not say that additional analysis of Bushman/O'Doherty data, or detailed reporting of your analysis of the resting vs activated sites from ref 47, are required to publish. However, I think it is important to let readers know that you have done some work in this regard, and that additional work is required to reach consensus. In this vein, please add the following as next to last paragraph of Discussion:

"An obvious consequence of our findings predict that RIGs would differ for HIV-1 in resting versus activated CD4+ T cells. Meta-analysis of these two classes of integration sites⁴⁷ however failed to distinguish such a difference, perhaps due to the presence of a few percent of cells expressing low levels of activation markers in the resting T cell set. Additional work will be required to assess comprehensively the RIGs that are used by HIV-1 in resting T cells."

Additional concerns:

1. Authors show that RIGs STAT5B and GRB2 do not translocate to the nuclear periphery when T cells are treated with JQ1 prior to activation (Fig S6G, H). At the same time, they show that JQ1 treatment does not significantly alter HIV-1 integration (Fig S2B-D). Don't these observations disprove the correlation of RIG peripheral nuclear localization and HIV-1 integration? The authors should comment on this disconnect in the Discussion.
2. The authors repeatedly indicate in rebuttal letters changes to the paper that reflect fair citation of published literature, but such changes do not always come through. Please further amend the paper as follows.
3. Intro lines 10-12 states HIV-1 does not readily infect resting T cells due to preintegration and integration blocks, citing 4 and 7. While true, it is only one part of the story. Una O'Doherty has clarified that resting cells do support a significant level of HIV-1 infection and integration, which, as discussed above, can reach within an order of magnitude of levels observed in activated cells. To more fairly present the field, please change "readily" to "efficiently" and also please include some citations from O'Doherty such as PMID: 19211752, 19493998, 22911005.
4. Page 3 line 10, rewrite: "regions of open chromatin, which in some studies map in proximity to the NPC."

5. Beginning of new section page 5, start sentence: "Some previous results showed that the majority"
6. Please add the following to line 2 of page 6: "These data are consistent with prior reports of similar integration patterns in cells as disparate as HeLa and CD34+ hemeatopoietic stem cells³⁶."
7. End of next to last paragraph page 7, please add: "We however do note that a separate study observed pan nuclear distribution of KDM2A and PACS1 in activated CD4+ T cells²³".
8. Page 8 line 8, please italicize STAT5B and GRB2.
9. Abstract line 11, since authors agree multiple forces are at play (SEs, IN-LEDGF, CA-CPSF6, etc), please delete the first "the" to read "along with their transcriptional activity are major determinants of"

Reviewer #3 (Remarks to the Author):

The paper has improved further with revision. The preferential mapping of integration sites to the A1 compartment of Jurkat T cells as assessed via novel Hi-C data is striking and carries the day. It will be interesting to see how these findings align with the works of others who do similar types of HIV integration site mappings. In this regard we are grateful the authors agree to release the sequences that define A1, A2, AB, B1 and B2 Jurkat cell regions.

We wish to thank the reviewer for helping us with improving the manuscript during the revision. Additional textual changes were inserted into the final version.

Personally, I am only partially swayed by the explanation provided in the rebuttal letter as to why careful analysis of the resting versus activated integration site data provided in reference 47 failed to show different RIG usage among the datasets. The authors are concerned that 3% of the resting cell population stains for markers of cell activation. At the same time, those authors highlighted that the levels of CD25, CD69, and HLA-DR on these 3% cells are “very low” and moreover that the resting cells did not detectably proliferate as assessed by BrdU incorporation, a gold standard in the field. I therefore feel we are getting into a debate of what truly are resting CD4 T cells and whose work is to be trusted in the field. Bushman and O’Doherty are generally well regarded and document integration as carefully as anyone; their follow up paper in PLoS P (PMID 22911005) carefully plots integration over time, where the resting cell level approaches within tenfold of the activated level by day 3. This paper also reported integration sites, so it may again be a source of material for the authors to query.

However I would not say that additional analysis of Bushman/O’Doherty data, or detailed reporting of your analysis of the resting vs activated sites from ref 47, are required to publish. However, I think it is important to let readers know that you have done some work in this regard, and that additional work is required to reach consensus. In this vein, please add the following as next to last paragraph of Discussion:

“An obvious consequence of our findings predict that RIGs would differ for HIV-1 in resting versus activated CD4+ T cells. While the meta-analysis of these two classes of integration sites failed to distinguish such a difference, perhaps due to the presence of a few percent of cells expressing low levels of activation markers in the resting T cell set. Additional work will be required to assess comprehensively the RIGs that are used by HIV-1 in resting T cells.”

The following paragraph has now been added to the discussion.

Additional concerns:

1. Authors show that RIGs STAT5B and GRB2 do not translocate to the nuclear periphery when T cells are treated with JQ1 prior to activation (Fig S6G, H). At the same time, they show that JQ1 treatment does not significantly alter HIV-1 integration (Fig S2B-D). Don’t these observations disprove the correlation of RIG peripheral nuclear localization and HIV-1 integration? The authors should comment on this disconnect in the Discussion.

In the experiment shown in FigS6G and 6H, *STAT5B* and *GRB2* loci did not change their position ie remained in the more interior regions of the nucleus. This effect was observed only if the cells were treated with JQ1 prior to their activation with CD3/CD28 beads, suggesting that loading of super-enhancers is required for the gene movement and genome reorganization in T cells. Thus, disassembly of super enhancers might compromise the peripheral movement/genome reorganization in T cells only if JQ1 treatment precedes T cell activation (with CD3/CD28).

Therefore, if the cells are first activated with CD3/CD28 and then treated with JQ1 no effect on gene positioning is observed (FISH data Figure S2D) and as a consequence HIV integration sites do not change (Figure S2B and S2C).

A comment on the role of super enhancers in gene organization and on the effect of JQ1 (observed only if treatment is done before T cell activation) was added to the discussion, at the end of the second paragraph on page 9.

2. The authors repeatedly indicate in rebuttal letters changes to the paper that reflect fair citation of published literature, but such changes do not always come through. Please further amend the paper as follows.

3. Intro lines 10-12 states HIV-1 does not readily infect resting T cells due to preintegration and integration blocks, citing 4 and 7. While true, it is only one part of the story. Una O'Doherty has clarified that resting cells do support a significant level of HIV-1 infection and integration, which, as discussed above, can reach within an order of magnitude of levels observed in activated cells. To more fairly present the field, please change "readily" to "efficiently" and also please include some citations from O'Doherty such as PMID: 19211752, 19493998, 22911005.

These references were not mentioned before. We have now added them to the manuscript.

4. Page 3 line 10, rewrite: "regions of open chromatin, which in some studies map in proximity to the NPC."

Changed as requested

5. Beginning of new section page 5, start sentence: "Some previous results showed that the majority"

We changed the sentence

6. Please add the following to line 2 of page 6: "These data are consistent with prior reports of similar integration patterns in cells as disparate as HeLa and CD34+ hemeatopoietic stem cells." In our work, we did not perform any comparative analysis between HeLa/CD34⁺ hematopoietic stem cell integration patterns and the ones in Jurkat and/or CD4⁺ T cells . Therefore, we would refrain from adding this conclusion to the result section.

7. End of next to last paragraph page 7, please add: "We however do note that a separate study observed pan nuclear distribution of KDM2A and PACS1 in activated CD4+ T cells".

8. Page 8 line 8, please italicize STAT5B and GRB2.

9. Abstract line 11, since authors agree multiple forces are at play (SEs, IN-LEDGF, CA-CPSF6, etc), please delete the first "the" to read "along with their transcriptional activity are major determinants of"

Points 7,8 and 9

All changes are inserted as requested.